# Lattice Boltzmann Model for Learning Real-World Pixel Dynamicity

**Guangze Zheng**[1], **Shijie Lin**[1], **Haobo Zuo**[1], **Si Si**[2], **Ming-Shan Wang**[2]
**Changhong Fu**[3], **Jia Pan**[1,4] [*]

[1]HKU    [2]Institute of Zoology, CAS    [3]Tongji University    [4]LimX Dynamics

guangze@connect.hku.hk, jpan@cs.hku.hk

Project page: https://george-zhuang.github.io/lbm

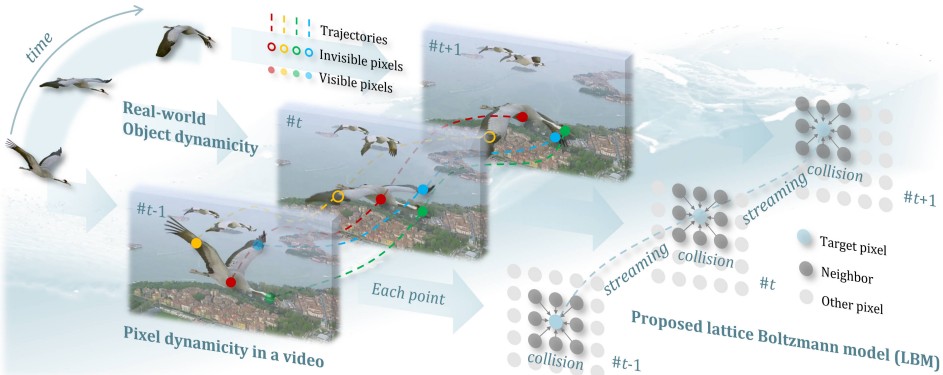

Figure 1: **Real-world object dynamicity → pixel dynamicity → fluid dynamicity**. The object dynamicity in the open world manifests through deformation and self-occlusion, as exemplified by the bird in the figure. From a visual perspective, such object dynamicity can be decomposed into pixel dynamicity. The pixels are subsequently modeled as fluid lattices that simulate hydrodynamic streaming and collision processes, and the pixel motion states are efficiently addressed with the proposed lattice Boltzmann model (LBM).

## Abstract

This work proposes the *Lattice Boltzmann Model* (**LBM**) to learn real-world pixel dynamicity for visual tracking. LBM decomposes visual representations into dynamic pixel lattices and solves pixel motion states through collision-streaming processes. Specifically, the high-dimensional distribution of the target pixels is acquired through a multilayer *predict-update* network to estimate the pixel positions and visibility. The *predict* stage formulates lattice collisions among the spatial neighborhood of target pixels and develops lattice streaming within the temporal visual context. The *update* stage rectifies the pixel distributions with online visual representations. Comprehensive evaluations of real-world point tracking benchmarks such as TAP-Vid and RoboTAP validate LBM's efficiency. A general evaluation of large-scale open-world object tracking benchmarks such as TAO, BFT, and OVT-B further demonstrates LBM's real-world practicality.

## 1 Introduction

Online and real-time pixel tracking is designed to achieve continuous localization of any specified pixel for real-world applications, such as embodied manipulation Wen et al. (2024); Zhang et al. (2024) and medical vision Schmidt et al. (2024). However, existing solutions are predominantly

---

[*]Corresponding author: jpan@cs.hku.hk

39th Conference on Neural Information Processing Systems (NeurIPS 2025).

offline Doersch et al. (2022); Cho et al. (2024) or semi-online Harley et al. (2022); Karaev et al. (2024b); Li et al. (2024b), leading to significant practical limitations: 1) *high resource consumption* from full video or time window buffering causes excessive memory usage, which is unsuitable for edge-device deployment in embodied systems; 2) *inevitable latency* due to the integrity of video or window input, preventing real-time inference; 3) *inadequate dynamic responsiveness*, lacking the ability to adapt to newly emerging pixels in videos immediately; 4) *privacy and storage concerns*, as storing full video data poses risks of privacy breaches and imposes substantial storage costs.

The limitations of offline and semi-online methods primarily stem from the inherent reliance on spatiotemporal integrity, which manifests in: 1) *temporal bi-directionality*, where point trajectories can be optimized through forward-backward analysis within the complete temporal context, 2) *iterative refinement*, involving multiple cost volume updates through identical networks to minimize estimation errors. Such reliance on spatiotemporal integrity compromises the real-time performance during inference. For instance, LocoTrack Cho et al. (2024) demonstrates an over 60% reduction of throughput from a single iteration to 4 iterations.

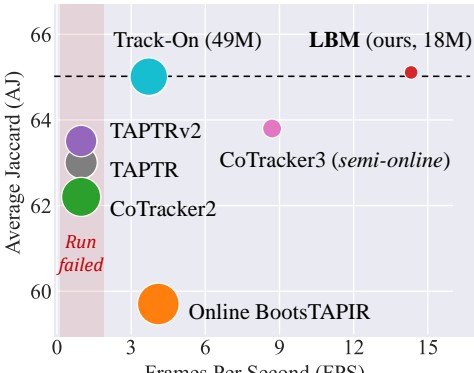

Figure 2: **Efficiency** comparison on TAP-Vid DAVIS benchmark with an NVIDIA Jetson Orin NX super. LBM shows efficiency with higher inference speed and smaller model size. The size of the circles corresponds to the number of parameters.

Before introducing the proposed LBM, we systematically outline its theoretical foundation in the lattice Boltzmann method Mohamad (2011) for fluid simulations. Fundamentally, the lattice Boltzmann method discretizes fluids into lattices where the distribution functions undergo collision and streaming operations governed by the Boltzmann transport equation. The inherent locality of collision and explicit time-stepping streaming contribute to computational efficiency.

Analogous to the lattice Boltzmann method, LBM discretizes the video into individual pixel lattices and estimates the motion states by characterizing the high-dimensional distributions of the lattices, as shown in Figure 1. Specifically, LBM employs a multi-layer *predict-update* network to estimate the distribution of specified pixel lattices. During the *predict* stage at each layer, the LBM employs collision and streaming operations to estimate the current target particle distribution. The collision step primarily accounts for the neighborhood distribution of lattices, modeling local lattice interactions. The streaming step governs the temporal evolution of the distribution function by propagating particles to adjacent lattices. In the *update* phase, the predicted lattice distribution is refined using online visual features. The position and visibility of target pixels at the current timestep are derived from the updated lattice distribution, processed through dedicated tracking and visibility heads. Compared with existing methods, LBM exhibits a distinct efficiency advantage, as illustrated in Figure 2.

To further address the dynamicity of real-world objects, LBM decomposes targets as ensembles of fine-grained pixels and establishes object associations through pixel tracking. Specifically, LBM decomposes object motion into more robust multi-pixel motion patterns, thereby enabling enhanced stability in resolving object kinematic states and stronger robustness against detection failures. In contrast to the method Zheng et al. (2024) that adopts window-based tracking, LBM dynamically prunes outlier pixels (*e.g.*, background and drifted pixels) and incorporates new inliers, thereby improving tracking responsiveness for highly dynamic objects.

## 2 Related Work

### 2.1 Tracking any point

**Offline and semi-online methods** The predominant frameworks for point tracking encompass offline methods that process the entire video and semi-online methods that rely on a multi-frame sliding window. PIPs Harley et al. (2022) and TAP-Net Doersch et al. (2022) establishes the point tracking baseline. TAPIR Doersch et al. (2023) and LocoTrack Cho et al. (2024) provide efficient solutions for cost volume computation. CoTracker series Karaev et al. (2024b,a) introduce proxy tokens to reduce computational cost. TAPTR Li et al. (2024b) and TAPTRv2 Li et al. (2024a) employ an architectural framework analogous to DETR Carion et al. (2020) and tracking points by detection.

Despite marked advancements in model performance, prevailing methods are still constrained to offline or window-based online paradigms that incur substantial systemic latency.

**Online methods** Driven by the pragmatic demands of real-world applications, online methods have witnessed a burgeoning emergence. MFT Neoral et al. (2024) extends the optical flow framework to multi-frame contexts. TAPIR Doersch et al. (2023)-related models achieve online adaptation through temporally causal masking. DynOMo Seidenschwarz et al. (2025) achieves online point tracking through dynamic 3D Gaussian reconstruction. Track-On Aydemir et al. (2025) further enhances online performance through spatiotemporal memory components. Compared to these works, the LBM places emphasis on tracking efficiency, particularly in real-time tracking under resource-constrained edge computing conditions, to meet the requirements of practical tracking applications.

## 2.2 Tracking dynamic objects

**Traditional object tracking methods** Multi-object tracking (MOT) predominantly focus on targets with limited dynamic characteristics in constrained scenarios, such as pedestrians Dendorfer et al. (2020) and vehicles Yu et al. (2020). Methods like TransTrack Sun et al. (2020), TrackFormer Meinhardt et al. (2022), and TransCenter Xu et al. (2022) adopt DETR Carion et al. (2020)-based architectures that model targets as learnable queries. However, these solutions typically represent targets as holistic entities and are vulnerable to performance degradation when handling highly dynamic targets. Such limitations become particularly pronounced during target deformation, partial occlusion, and fast motion.

**Open-world object tracking methods** Recent advancements have extended MOT to diverse scenarios and arbitrary object categories. extend to diverse scenarios and arbitrary targets. OVTrack Li et al. (2023) and MASA Li et al. (2024c) integrate text encoder to specify tracking targets. UNINEXT Yan et al. (2023) and GLEE Wu et al. (2024) adapt open-world detection architectures to tracking tasks through fine-tuning on videos. Motion modeling methods from conventional trackers like SORT Bewley et al. (2016) can achieve open-world tracking through integration with open-vocabulary detectors. NetTrack Zheng et al. (2024) addresses dynamic targets through decomposition of holistic objects into nets, enabling fine-grained tracking. Considering the vulnerability of most methods to catastrophic detection failures in applications, LBM adopts fine-grained pixel tracking and high-responsive updates to ensure robust applicability across diverse dynamic scenarios.

## 3 Method

### 3.1 Preliminary: lattice Boltzmann method

The lattice Boltzmann method solves the discrete velocity of the fluid on the lattice with streaming and collision processes. Given position $\mathbf{x}$ and time $t$, the discrete distribution function $\mathbf{f}$ is as follows:

$$\mathbf{f}(\mathbf{x}, t) = \sum_i [f_i(\mathbf{x} - \mathbf{c}_i \Delta t, t - \Delta t) + \Omega_i(\mathbf{x} - \mathbf{c}_i \Delta t, t - \Delta t)], \tag{1}$$

where $\Delta t$ and $\mathbf{c}_i$ denote the time step and discrete velocity in the $i$-th direction, respectively. $\Omega$ describes the collision of lattices on each node, which describes the relaxation of the distribution function towards the equilibrium distribution. By solving for streaming-collision processes, the density $\rho$ and discrete velocity $\mathbf{u}$ of the fluid are obtained as:

$$\rho(\mathbf{x}, t) = \sum_i f_i(\mathbf{x}, t), \quad \rho\mathbf{u}(\mathbf{x}, t) = \sum_i \mathbf{c}_i f_i(\mathbf{x}, t). \tag{2}$$

### 3.2 Lattice Boltzmann model for point tracking

Given the input image $\mathbf{I} \in \mathbb{R}^{3 \times H \times W}$ and $N$ query points $\mathbf{q} \in \mathbb{R}^{N \times 2}$, lattice Boltzmann model (LBM) estimates the positions $\mathbf{p} \in \mathbb{R}^{N \times 2}$ and visibility $\mathbf{v} \in \mathbb{R}^N$ of these points in subsequent image streams in the real-time and online manner. This process is achieved by treating the query points as fluid particles and solving their $d$-dimensional distribution $\mathbf{f} \in \mathbb{R}^{N \times d}$ in dynamic scenes through streaming and collision processes. As shown in Figure 3, the specific steps include the following:

**Visual encoding** To model visual representations, LBM employs the first three layers of a ResNet18 He et al. (2016) model pre-trained on ImageNet Deng et al. (2009) as the visual encoder. In contrast to previous methods that process multi-level features separately, we follow Xie et al.

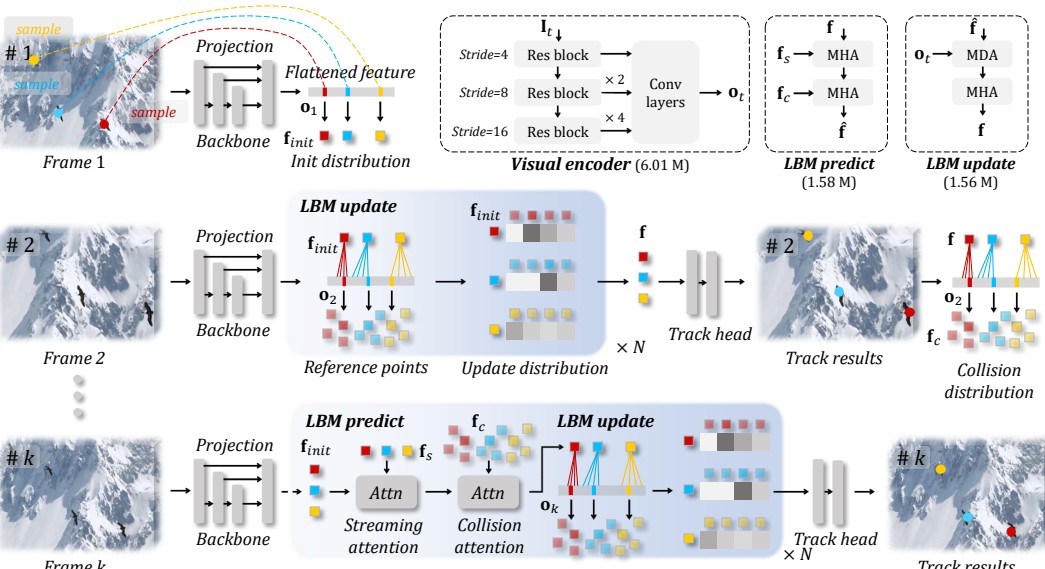

Figure 3: **LBM framework for point tracking**, illustrating 1) the distribution initialization process in LBM, 2) the *LBM update* step that incorporates online visual features to update the pixel distribution, 3) the derivation of streaming and collision distributions, and 4) the *LBM predict* process that utilizes both streaming and collision distributions to predict current pixel distribution. Lightweight architectures have been implemented for LBM modules to accommodate real-world deployment requirements.

(2021) by upsampling all feature maps to a stride of 4 and concatenating them after projections, and acquire the output visual representations $\mathbf{o} \in \mathbb{R}^{d \times \frac{H}{4} \times \frac{W}{4}}$. The design of the visual encoder primarily considers efficiency.

**Distribution initialization** It is essential to initialize the distribution function when formulating query points as fluid particles. LBM accomplishes this by sampling the visual representations $\mathbf{o}$ corresponding to the query points $\mathbf{q}$, *i.e.*, $\mathbf{f}_{init} = \text{BilinearSample}(\mathbf{o}, \mathbf{q}) \in \mathbb{R}^{N \times d}$.

**Distribution prediction** Corresponding to Equation 1, within a new time step, LBM learns from the previous distribution functions and predicts the distribution functions at the current moment $t$ through the streaming and collision processes. Differently, LBM consolidates the distribution functions from multiple directions into a single $d$-dimensional distribution. In contrast to Equation 1, LBM does not employ fixed neighboring pixels as collision elements. Instead, it computes the interaction between the pixel distribution and a learnable neighborhood $\delta$, thereby ensuring adaptability to dynamic scenes. At this stage, the prediction step can be formulated as follows.

$$\mathbf{f}(x, t|\delta) = \mathbf{f}(x, t - \Delta t|\delta) + \Omega(x, t - \Delta t|\delta) , \qquad (3)$$

where the collision operator $\Omega$ is implemented as the deformable attention Zhu et al. (2020). For stronger robustness, the temporal context is further extended from a single historical time step to $N_s$, consisting of streaming distributions $\mathbf{f}_s \in \mathbb{R}^{N \times N_s \times d}$ and collision distributions $\mathbf{f}_c \in \mathbb{R}^{N \times N_s \times d}$. To ensure the stability of pixel distributions, we initialize the distribution with $\mathbf{f}_{init}$ at each time step and facilitate its interaction with $\mathbf{f}_s$ and $\mathbf{f}_c$ via cross-attention modules $\phi$. Equation 3 is reformulated as:

$$\mathbf{f} = \phi_c(\phi_s(\mathbf{f}_{init}, \mathbf{f}_s), \mathbf{f}_c) , \quad \mathbf{f}_s = \{\mathbf{f}_i\}_{i=t-N_s}^{t-1} , \quad \mathbf{f}_c = \{\Omega(\mathbf{f}_i, \mathbf{o}_i, \mathbf{p}_i|\delta_i)\}_{i=t-N_s}^{t-1} . \qquad (4)$$

The details of the collision process are discussed in the Appendix A.1.

**Distribution update** In a new time step $t$ and its corresponding image $\mathbf{I}_t$, pixels should dynamically update their positions $\mathbf{p}$ and distribution functions $\mathbf{f}$. LBM first computes the correlation map between the pixel distribution $\mathbf{f}$ and visual representations $\mathbf{o}$. The top-$k$ response values from this map are then selected as reference points $\mathbf{r} \in \mathbb{R}^{N \times k \times 2}$ to update the pixel distribution function via deformable attention module $\psi$ as: $\psi(\mathbf{f}, \mathbf{o}, \mathbf{r})$. The update stage primarily refines the distribution function by integrating visual representations from multiple latent potential positions. The adoption of deformable attention remains instrumental in enhancing computational efficiency.

**Multi-layer predict-update Transformer** Different from approaches employing multiple iterations, LBM employs a multi-layer Transformer architecture, which enhances inference efficiency while

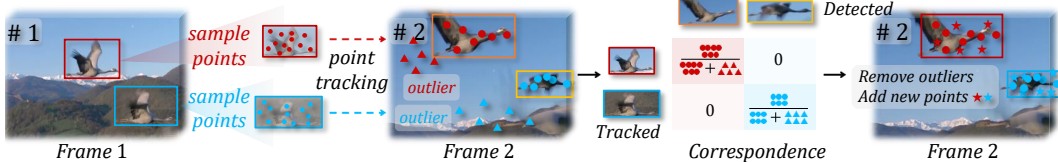

Figure 4: **LBM framework for object tracking.** LBM is initialized by sampling multiple fine-grained pixels within the target boxes, matches with pixels, and dynamically prunes outliers while replenishing with newly sampled inliers.

maintaining high tracking accuracy. Each Transformer layer comprises a prediction step and an update step as discussed earlier. As the depth of the Transformer layers increases, the number of reference points in the update stage progressively decreases layer by layer, and is ultimately reduced to one in the last layer, serving as the definitive reference point $\mathbf{r}_{last} \in \mathbb{R}^{N \times 2}$.

The final output distribution functions are fed into the track head and visibility head to predict the point position $\mathbf{p}$ and visibility $\mathbf{v}$, respectively. Corresponding to Equation 2, the track head predicts the offset $\Delta \mathbf{p} \in \mathbb{R}^{N \times 2}$ of tracked points from the final reference points $\mathbf{r}_{last}$. Following previous work Karaev et al. (2024a), the confidence $\rho \in \mathbb{R}^N$ and visibility $\mathbf{v} \in \mathbb{R}^N$ are predicted through a coupled head. These processes are as follows:

$$\Delta \mathbf{p} = \mathcal{H}_{track}(\mathbf{f}, \mathbf{o}, \mathbf{r}_{last}), \quad \{\rho, \mathbf{v}\} = \mathcal{H}_{vis}(\mathbf{f}, \mathbf{o}, \mathbf{r}_{last}). \tag{5}$$

Both heads $\mathcal{H}_{track}$ and $\mathcal{H}_{vis}$ consist of a deformable attention module and an MLP layer.

**Loss** The composition of the loss in LBM is as follows:

$$\mathcal{L} = \lambda_{cls}\mathcal{L}_{cls} + \mathcal{L}_{reg} + \mathcal{L}_{vis} + \mathcal{L}_{conf} \quad, \tag{6}$$

The cross-entropy loss is employed to the correlation map $\mathbf{c}$ at each layer of the Transformer as $\mathcal{L}_{cls} = \text{CE}(\mathbf{c}, \mathbf{c}_{gt}|\mathbf{v}_{gt})$. The offset $\Delta \mathbf{p}$ is supervised by L1 loss as $\mathcal{L}_{reg} = \text{L1}(\Delta \mathbf{p}, \Delta \mathbf{p}_{gt}|\mathbf{v}_{gt})$. Only visible points are considered in the above two losses. The visibility loss and confidence loss both adopt cross-entropy losses as:$\mathcal{L}_{vis} = \text{CE}(\sigma(\mathbf{v}), \mathbf{v}_{gt})$, $\mathcal{L}_{conf} = \text{CE}(\sigma(\rho), \mathbb{1}[\|\mathbf{p} - \mathbf{p}_{gt}\| < 8])$. Please refer to Appendix A.1 for details.

### 3.3 Lattice Boltzmann model for object tracking

In object tracking, LBM takes image $\mathbf{I} \in \mathbb{R}^{3 \times H \times W}$ and corresponding $M$ detection boxes $\mathbf{b} \in \mathbb{R}^{M \times 4}$ as input, and associates objects in the subsequent image streams.

**Matching** Associating instances across consecutive time steps is realized by point-based matching, as shown in Figure 4. Compared with Zheng et al. (2024), LBM demonstrates higher efficiency by eliminating prior coarse matching within temporal windows. Specifically, upon receiving a new instance, $N$ inlier pixels are randomly sampled from the bounding box as fine-grained pixels and initialized. As a new frame arrives, the LBM predicts both positional coordinates and visibility states of the pixels. Instance association is subsequently achieved by evaluating the spatial correspondence between the predicted pixels and the inlier pixels within the bounding boxes of instances in the new frame. Please refer to Appendix A.2 for implementation details.

**Update** LBM's real-time responsiveness further benefits object tracking by eliminating background pixels, thereby enhancing robustness under noise. Specifically, LBM implements dynamic point management: pixels persistently residing outside target bounding boxes for consecutive frames are systematically eliminated during each update cycle. Concurrently, novel inlier pixels are replenished within the current bounding box. The update mechanism effectively maintains instance representation integrity while accommodating challenging target dynamicity, including partial occlusions, deformations, and background changes that usually cause tracking failures.

## 4 Experiments

### 4.1 Experimental setup

The critical details of the experimental setup are discussed as follows. Please refer to Appendix B for more training and evaluation details.

Table 1: **Real-world point tracking performance** on TAP-Vid DAVIS, TAP-Vid Kinetics, and RoboTAP datasets. ∗ denotes training on additional data. LBM reaches SOTA online performance with fewer parameters and higher efficiency, even compared with offline methods.

| Model | Params | TAP-Vid DAVIS | | | TAP-Vid Kinetics | | | RoboTAP | | |
|---|---|---|---|---|---|---|---|---|---|---|
| | | AJ↑ | $\delta^x_{avg}$↑ | OA↑ | AJ↑ | $\delta^x_{avg}$↑ | OA↑ | AJ↑ | $\delta^x_{avg}$↑ | OA↑ |
| *1) Offline* | | | | | | | | | | |
| TAPIR Doersch et al. (2023) | 31M | 56.2 | 70.0 | 86.5 | 49.6 | 64.2 | 85.0 | 59.6 | 73.4 | 87.0 |
| LocoTrack Cho et al. (2024) | 12M | 62.9 | 75.3 | 87.2 | 52.9 | 66.8 | 85.3 | 62.3 | 76.2 | 87.1 |
| BootsTAPIR* Doersch et al. (2024) | 78M | 61.4 | 73.6 | 88.7 | 54.6 | 68.4 | 86.5 | 64.9 | 80.1 | 86.3 |
| CoTracker3 Karaev et al. (2024a) | 25M | 63.3 | 76.2 | 88.0 | 53.5 | 66.5 | 86.4 | 59.9 | 73.4 | 87.1 |
| CoTracker3* Karaev et al. (2024a) | 25M | 64.4 | 76.9 | 91.2 | 54.7 | 67.8 | 87.4 | 64.7 | 78.8 | 90.8 |
| *2) Window-based Online* | | | | | | | | | | |
| PIPs Harley et al. (2022) | 29M | 42.2 | 64.8 | 77.7 | 31.7 | 53.7 | 72.9 | - | - | - |
| PIPs++ Zheng et al. (2023) | 25M | - | 73.7 | - | - | 63.5 | - | - | 63.0 | - |
| CoTracker Karaev et al. (2024b) | 45M | 61.8 | 76.1 | 88.3 | 49.6 | 64.3 | 83.3 | 58.6 | 73.4 | 87.0 |
| TAPTR Li et al. (2024b) | 42M | 63.0 | 76.1 | 91.1 | 49.0 | 64.4 | 85.2 | 60.1 | 75.3 | 86.9 |
| TAPTRv2 Li et al. (2024a) | 41M | 63.5 | 75.9 | 91.4 | 49.7 | 64.2 | 85.7 | 60.9 | 74.6 | 87.7 |
| SpatialTracker Xiao et al. (2024) | 34M | 61.1 | 76.3 | 89.5 | 50.1 | 65.9 | 86.9 | - | - | - |
| CoTracker3 Karaev et al. (2024a) | 25M | 64.5 | 76.7 | 89.7 | 54.1 | 66.6 | 87.1 | 60.8 | 73.7 | 87.1 |
| CoTracker3* Karaev et al. (2024a) | 25M | 63.8 | 76.3 | 90.2 | 55.8 | 68.5 | 88.3 | 64.7 | 78.0 | 89.4 |
| *3) Online* | | | | | | | | | | |
| DynOMo Seidenschwarz et al. (2025) | - | 45.8 | 63.1 | 81.1 | - | - | - | - | - | - |
| MFT Neoral et al. (2024) | - | 47.3 | 66.8 | 77.8 | 39.6 | 60.4 | 72.7 | - | - | - |
| Online TAPIR Doersch et al. (2023) | 31M | 56.7 | 70.2 | 85.7 | 51.5 | 64.4 | 85.2 | 59.1 | - | - |
| DOT Le Moing et al. (2024) | - | 53.5 | 67.8 | 85.4 | 45.3 | 58.0 | 81.4 | 51.9 | 62.9 | 79.9 |
| Track-On Aydemir et al. (2025) | 49M | 65.0 | **78.0** | **90.8** | 53.9 | **67.3** | **87.8** | **63.5** | **76.4** | **89.4** |
| **LBM** (ours) | **18M** | **65.1** | 77.5 | 89.5 | 53.4 | 66.9 | 86.1 | 61.4 | 75.8 | 87.4 |

**Training** Consistent with previous works Doersch et al. (2022); Karaev et al. (2024b), LBM uses TAP-Vid Kubric Greff et al. (2022) dataset for training, which contains $11k$ video sequences of 24 frames each. The training process encompasses 150 epochs (approximately $100k$ iterations) using 4 NVIDIA H800 GPUs with a total batch size of 16 and FP16 mixed-precision.

**Evaluation datasets** Three real-world point tracking benchmarks are employed, including TAP-Vid DAVIS, TAP-Vid Kinetics, and RoboTAP Vecerik et al. (2024). Open-world object tracking datasets include: TAO Dave et al. (2020) validation set, BFT Zheng et al. (2024) test set, and OVT-B Liang and Han (2024).

**Evaluation metrics** For point tracking, evaluation adheres to TAP-Vid benchmark Doersch et al. (2022), comprising average Jaccard (AJ), $\delta^x_{avg}$, and occlusion accuracy (OA). The evaluation metrics for open-world object tracking include TETA Li et al. (2022) and OWTA Liu et al. (2022). TETA is a comprehensive metric assessing association accuracy (AssA), localization accuracy (LocA), and classification accuracy (ClsA). The object categories are divided into *novel* and *base*.

## 4.2 Main results

**Point tracking** performance evaluation is summarized in Table 1. State-of-the-art (SOTA) methods are categorized into three classes: *offline*, *window-based online*, and *online*. *Offline* methods ingest full video sequences as input, *window-based* approaches process temporal segments of 8 or 16 frames Karaev et al. (2024b), while *online* methods exclusively utilize the current frame with per-frame inference, achieving optimal responsiveness and demonstrating strong practicality for real-world deployment. LBM achieves SOTA performance with an exceptionally lean parameter configuration (18 M), surpassing most existing window-based online and offline methods. Notably, LBM reaches the SOTA performance with only 37% parameters compared with Track-On. As quantitatively validated in Figure 2, LBM demonstrates real-time operational capability at 14.3 FPS on the NVIDIA Jetson Orin NX Super edge platform, with a 3.9× speed advantage over Track-On, thereby showing computational efficiency and practicality in real-world environments.

**Object tracking** performance of LBM and other SOTA methods is systematically compared in Table 2 for TAO, Table 3 for OVT-B, and Table 4 for BFT. The evaluated models are categorized into two paradigms based on training strategies: *additional training* with trackers fine-tuned with

Table 2: **Real-world object tracking performance** on TAO validation dataset. Without training on domain-specific data of object tracking, LBM demonstrates state-of-the-art performance.

| Model | All | | | | Base | | | | Novel | | | |
|---|---|---|---|---|---|---|---|---|---|---|---|---|
| | TETA↑ | LocA↑ | AssocA↑ | ClsA↑ | TETA↑ | LocA↑ | AssocA↑ | ClsA↑ | TETA↑ | LocA↑ | AssocA↑ | ClsA↑ |
| *1) Additional training* | | | | | | | | | | | | |
| Tractor++ Bergmann et al. (2019) | 28.0 | 49.0 | 22.8 | 12.1 | 28.3 | 47.4 | 20.5 | 17.0 | 22.7 | 46.7 | 19.3 | 2.2 |
| DeepSORT Wojke et al. (2017) | 26.0 | 48.4 | 17.5 | 12.1 | 26.9 | 47.1 | 15.8 | 17.7 | 21.1 | 46.4 | 14.7 | 2.3 |
| UNINEXT Yan et al. (2023) | 31.9 | 43.4 | 35.5 | 17.1 | - | - | - | - | - | - | - | - |
| AOA Du et al. (2021) | 25.3 | 23.4 | 30.6 | 21.9 | - | - | - | - | - | - | - | - |
| QDTrack Pang et al. (2021) | 30.0 | 50.5 | 27.4 | 12.1 | 27.1 | 45.6 | 24.7 | 11.0 | 22.5 | 42.7 | 24.4 | 0.4 |
| TETer Li et al. (2022) | 40.1 | 56.3 | 39.9 | 24.1 | - | - | - | - | - | - | - | - |
| OVTrack Li et al. (2023) | 34.7 | 49.3 | 36.7 | 18.1 | 35.5 | 49.3 | 36.9 | 20.2 | 27.8 | 48.8 | 33.6 | 1.5 |
| GLEE-Plus Wu et al. (2024) | 41.5 | 52.9 | 40.9 | **30.8** | - | - | - | - | - | - | - | - |
| MASA Li et al. (2024c) | **46.3** | **65.8** | **44.1** | 28.9 | **47.0** | **66.0** | 44.5 | 30.5 | **40.8** | 64.4 | **41.2** | **17.0** |
| SLAck Li et al. (2024d) | 41.1 | 56.3 | 41.8 | 25.1 | - | - | - | - | - | - | - | - |
| OVTrack+ Liang and Han (2024) | 38.4 | 57.5 | 40.8 | 16.9 | 39.2 | 57.5 | 41.0 | 18.9 | 32.5 | 57.0 | 38.7 | 1.8 |
| *2) Training-free* | | | | | | | | | | | | |
| SORT Bewley et al. (2016) | 24.9 | 48.1 | 14.3 | 12.1 | - | - | - | - | - | - | - | - |
| Tractor Bergmann et al. (2019) | 24.2 | 47.4 | 13.0 | 12.1 | - | - | - | - | - | - | - | - |
| ByteTrack Zhang et al. (2022) | 27.6 | 48.3 | 20.2 | 14.4 | 28.2 | 50.4 | 18.1 | 16.0 | 22.0 | 48.2 | 16.6 | 1.0 |
| OC-SORT Cao et al. (2023) | 28.6 | 49.7 | 21.8 | 14.3 | 28.9 | 51.4 | 19.8 | 15.4 | 23.7 | 49.6 | 20.4 | 1.1 |
| NetTrack Zheng et al. (2024) | - | - | - | - | 33.0 | 45.7 | 28.6 | 24.8 | 32.6 | 51.3 | **33.0** | **13.3** |
| **LBM** (ours) | **45.3** | **70.0** | **32.4** | **33.4** | **46.5** | **69.9** | **33.2** | **36.4** | **36.1** | **70.8** | 26.2 | 11.4 |

Table 3: **Real-world object tracking performance** on OVT-B dataset. Without training on domain-specific data of object tracking, LBM demonstrates state-of-the-art performance.

| Model | All | | | | Base | | | | Novel | | | |
|---|---|---|---|---|---|---|---|---|---|---|---|---|
| | TETA↑ | LocA↑ | AssocA↑ | ClsA↑ | TETA↑ | LocA↑ | AssocA↑ | ClsA↑ | TETA↑ | LocA↑ | AssocA↑ | ClsA↑ |
| *1) Additional training* | | | | | | | | | | | | |
| OVTrack Li et al. (2023) | 46.8 | 60.5 | 66.7 | **13.4** | 45.5 | 61.1 | 65.5 | **9.6** | 46.1 | 60.8 | 66.1 | **11.5** |
| OVTrack+ Liang and Han (2024) | **47.6** | **61.6** | **68.2** | 13.2 | **46.4** | **62.5** | **67.3** | 9.4 | **47.0** | **62.0** | **67.7** | 11.3 |
| *2) Training-free* | | | | | | | | | | | | |
| ByteTrack Zhang et al. (2022) | 20.6 | 35.6 | 12.7 | 13.4 | 19.6 | 36.6 | 12.0 | 10.3 | 20.1 | 36.1 | 12.4 | 11.9 |
| OC-SORT Cao et al. (2023) | 16.5 | 31.0 | 4.4 | 14.3 | 15.4 | 31.4 | 4.3 | 10.3 | 16.0 | 31.2 | 4.3 | 12.3 |
| StrongSORT Du et al. (2023) | 25.7 | 31.4 | 31.6 | 14.2 | 23.9 | 31.8 | 29.7 | 10.3 | 24.8 | 31.6 | 30.7 | 12.2 |
| **LBM** (ours) | **56.8** | **75.7** | **71.7** | **22.9** | **57.5** | **74.7** | **72.4** | **25.5** | **56.0** | **76.7** | **70.9** | **20.3** |

supplementary data and tracking annotations, *e.g.*, TAO training set, YTVIS Yang et al. (2019)), and *training-free* with models operating without leveraging domain-specific tracking supervision. LBM establishes SOTA performance and demonstrates methodological universality, outperforming both non-finetuned approaches and domain-specific finetuning strategies. On the TAO benchmark, LBM achieves comparable performance to MASA. Notably, when processing identical detection inputs as GLEE-Plus, LBM delivers statistically significant +4.2 gains on TETA.

Table 4: **Real-world dynamic object tracking performance** on BFT dataset.

| Model | OWTA↑ | D. Re.↑ | A. Acc.↑ |
|---|---|---|---|
| *1) Finetuned on BFT train set* | | | |
| CenterTrack Zhou et al. (2020) | 61.6 | 70.5 | 54.0 |
| FairMOT Zhang et al. (2021) | 40.2 | 57.5 | 28.2 |
| TransTrack Sun et al. (2020) | 66.8 | 73.9 | 60.3 |
| TrackFormer Meinhardt et al. (2022) | **67.4** | **74.5** | **61.1** |
| TransCenter Xu et al. (2022) | 63.5 | 73.2 | 55.3 |
| *2) Zero-shot setting* | | | |
| StrongSORT Du et al. (2023) | 43.2 | 54.7 | 34.2 |
| SORT Bewley et al. (2016) | 59.9 | 63.9 | 56.2 |
| IOUTracker Bochinski et al. (2017) | 70.9 | 77.4 | 65.0 |
| ByteTrack Zhang et al. (2022) | 64.1 | 67.9 | 60.5 |
| OC-SORT Cao et al. (2023) | 69.0 | 70.9 | 67.2 |
| NetTrack Zheng et al. (2024) | 72.5 | **80.7** | 65.2 |
| **LBM** (ours) | **74.5** | 80.0 | **69.4** |

LBM achieves best performance on the OVT-B benchmark with a +9.2 TETA gain over SOTA OVTrack+. These cross-domain advancements substantiate LBM's zero-shot generalization capacity without dataset-specific adaptation. Further validating operational robustness, LBM attains 74.5 OWTA on the BFT benchmark to track highly dynamic objects, surpassing NetTrack by a +2.0 OWTA gain, demonstrating the ability to track highly dynamic objects.

## 4.3  Ablation study

**Collision and streaming** Module ablation is shown in Table 5, which shows 1) removing the streaming module eliminated historical distribution, thereby erasing temporal context and making it susceptible to abrupt changes in dynamic pixel distributions, resulting in a 0.9 AJ degradation; 2) disabling the collision module deprived pixels of neighborhood distribution, causing locality constraints. Compared to streaming module removal, this incurred 1.2 OA reduction, indicating heightened vulnerability to occlusions; 3) both modules contributed to performance gains at the cost of increased parameters and approximately 11 ms latency on the NVIDIA Jetson Orin NX.

Table 5: **Module ablation of LBM** on TAP-Vid DAVIS benchmark. The speeds are tested on an NVIDIA Jetson Orin NX super.

| Modules | | | | | | | | | | | |
|---|---|---|---|---|---|---|---|---|---|---|---|
| Streaming | Collision | Params | Speed↑ | AJ↑ | $\delta_{avg}^x$↑ | OA↑ | $\delta^{1px}$↑ | $\delta^{2px}$↑ | $\delta^{4px}$↑ | $\delta^{8px}$↑ | $\delta^{16px}$↑ |
| ✓ | ✓ | 17.8 M | 14.3 FPS | 65.1 | 77.5 | 89.5 | 46.8 | 70.2 | 84.9 | 91.3 | 94.6 |
| ✗ | ✓ | 15.4 M | 17.1 FPS | 64.2 | 77.0 | 89.2 | 46.0 | 69.7 | 84.7 | 90.7 | 94.1 |
| ✓ | ✗ | 14.7 M | 17.8 FPS | 63.6 | 76.9 | 88.3 | 46.4 | 69.7 | 83.9 | 90.6 | 93.9 |
| ✗ | ✗ | 12.4 M | 21.5 FPS | 51.8 | 66.2 | 77.0 | 40.1 | 59.5 | 71.3 | 77.5 | 82.6 |

Table 6: **Ablation on number of predict-update layers**. 3 predict-update layers achieve better efficiency on TAP-Vid DAVIS.

| $N_{layer}$ | Params | Speed↑ | AJ↑ | $\delta_{avg}^x$↑ | OA↑ |
|---|---|---|---|---|---|
| 2 | 14.7 M | 18.5 FPS | 64.2 | 77.3 | 89.3 |
| 3 | 17.8 M | 14.3 FPS | 65.1 | 77.5 | 89.5 |
| 4 | 21.0 M | 11.2 FPS | 64.9 | 77.3 | 89.6 |

Table 7: **Ablation on the visual encoder**. ResNet18 obtains better efficiency compared with Swin-T on TAP-Vid DAVIS.

| Encoder | Params | Speed↑ | AJ↑ | $\delta_{avg}^x$↑ | OA↑ |
|---|---|---|---|---|---|
| Swin-T | 27.3 M | 8.9 FPS | 63.0 | 76.6 | 88.7 |
| ResNet18 | 17.8 M | 14.3 FPS | 65.1 | 77.5 | 89.5 |

**Transformer layer** The number of predict-update layers in the Transformer is discussed in Table 6. Each predict-update layer contains approximately 3.1 M parameters and introduces a computational latency of 18 ms. Compared to the 2-layer architecture, the 3-layer model demonstrates a performance gain of +0.9 AJ metric. However, the 4-layer configuration shows negligible improvement over its 3-layer counterpart, indicating the existence of performance saturation in deeper network configurations for point tracking. Therefore, LBM adopts the 3-layer configuration as the default architectural setting, achieving an optimal balance between computational efficiency and model performance.

**Visual encoder** As shown in Table 7, we substituted ResNet18 with Swin-T Liu et al. (2021) while maintaining identical implementation protocols: utilizing ImageNet pre-trained weights, extracting hierarchical features from blocks with stride configurations of [4, 8, 16], and spatially aligning these multi-scale representations through convolutional projection layers to stride=4 followed by channel-wise concatenation. Despite introducing 9.5 M additional parameters and incurring a 42 ms computational overhead, the architectural substitution demonstrated a 2.1 AJ metric degradation compared to the ResNet18 baseline, highlighting the non-trivial performance trade-offs. The observed performance discrepancy could be attributed to ResNet's superior capability in preserving spatial integrity, particularly through enhanced spatial alignment during hierarchical

**Spatial awareness** CoTracker proposes expanding the number of queries to enhance the model's spatial awareness. Therefore, Track-On, CoTracker3, and CoTracker employ initialized $K \times K$ grid points as extended queries. However, increasing query number typically compromises the model's inference speed, especially in real-world applications. In contrast, LBM benefits from learning the pixel collision process and inherently possesses stronger spatial perception capabilities, as illustrated in Figure 5. Without additional extended queries, the performance improvement of LBM is less pronounced compared to other methods (+0.1 against -1.7, -1.6, and -1.8 on AJ metric), demonstrating its better spatial awareness.

**Dynamic object tracking** on BFT benchmark is shown in Figure 6. The tolerance timeout denotes the maximum frames allowed for allocated pixels to be outliers. A higher timeout slows pixel updates, reduces fine-grained dynamicity, and hence degrades tracking performance. A timeout of 1 frame induces excessively low tolerance and influences the stability of fine-grained pixels. Therefore, a

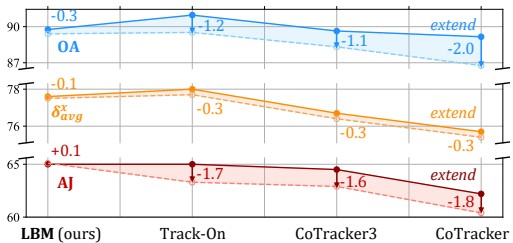

Figure 5: **Ablation on extended queries**. LBM benefits from spatial awareness and reduces dependency on extended queries for efficiency.

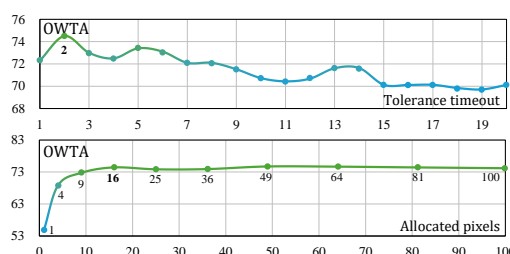

Figure 6: **Ablation on LBM for object tracking**. The tolerance timeout and number of allocated pixels are taken into account.

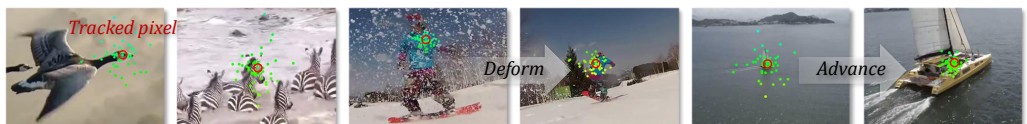

Figure 7: **Dynamic neighbor visualization**. A single pixel is tracked, and its neighbors are visualized. The tracked pixels are marked by red circles.

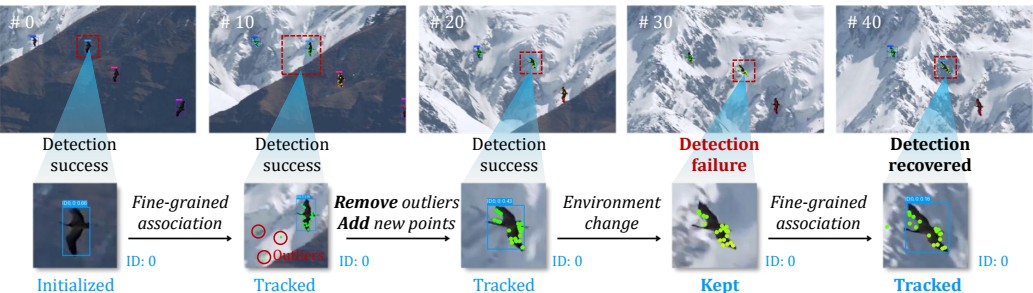

Figure 8: **Object tracking visualization**. LBM achieves robust object tracking by learning the dynamic pixel trajectories of the object, effectively mitigating the issue of detection failure.

timeout of 2 frames is set by default. Furthermore, tracking performance improves with increased pixel allocation per object, but plateaus beyond 16 pixels.

## 4.4 Efficiency

Efficiency comparison on an NVIDIA Jetson Orin NX super (16 GB) is shown in Figure 2. The speed of *online* and *semi-online* models is evaluated on TAP-Vid DAVIS, and the runs of TAPTRv2, TAPTR, and CoTracker2 fail due to insufficient resources. LBM demonstrates higher efficiency due to its lightweight architecture and inference speed. Although CoTracker3 adopts a window-based online structure capable of processing 16 images in a single pass, its inference speed remains significantly slower than LBM under resource constraints of edge devices, while still suffering from semi-online latency. None of the models employs extended queries in this comparison for efficiency.

**TensorRT quantization** To further mitigate deployment complexity in real-world applications, LBM is compiled into an ONNX format to enhance cross-platform deployment compatibility. Furthermore, FP16 quantization via TensorRT was implemented to accelerate inference on widely adopted embedded systems such as the NVIDIA Jetson series. After quantization, the quantized model realizes a ×3.5 acceleration with 49 FPS.

## 4.5 Visualization

**Dynamic neighbors** Benefiting from multi-scale deformable attention, the tracked pixels learn from dynamic neighboring regions during collision process, thereby enhancing spatial perception capabilities. The dynamic neighbors are visualized in Figure 7. This enables LBM to maintain robust target tracking even when dynamic objects undergo deformation and fast motion. As the camera advances toward the target, the enlarged target scale stabilizes the appearance of tracked pixels, while observable dynamic neighbors demonstrate enhanced spatial aggregation characteristics and concentrate on adjacent spatial regions.

**Tracking against detection failure** As shown in Figure 8, LBM demonstrates tracking robustness against detection failures. During initialization, the object detection yields favorable results, and fine-grained pixels are sampled from the bounding box. Over the subsequent frames, the fine-grained pixels are tracked, with their trajectories utilized for association. During this process, pixels that extend beyond the bounding box are removed, while new pixels are sampled within the bounding box. When detection fails due to environmental changes, the tracking method typically also fails, which is detrimental to real-world applications. In contrast, LBM can persistently track the fine-grained pixels of the object, maintaining robust tracking once detection recovers. The detection results are provided by YOLOE-11-L Wang et al. (2025).

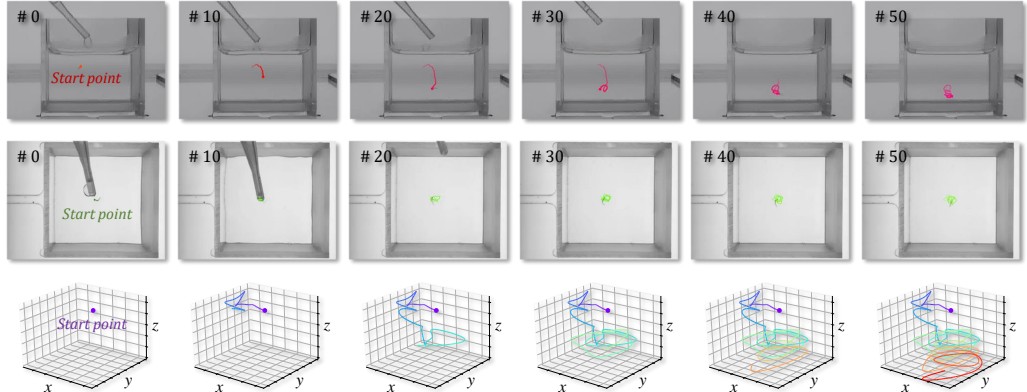

Figure 9: **LBM in real-world applications: behavioral analysis of zebrafish**. Given two independent multi-view videos, LBM enables three-dimensional trajectory reconstruction of zebrafish, facilitating quantitative behavioral analysis.

## 4.6 Real-world application

Figure 9 shows the behavioral analysis of zebrafish with target gene knockout. Utilizing two orthogonal perspectives (top and lateral views), the LBM enables researchers to reconstruct three-dimensional trajectories of zebrafish swimming behavior induced by pipette transfer into a container. The quantitative analysis revealed that with the specific gene knockout, zebrafish exhibited pronounced rotational swimming patterns, thereby demonstrating LBM's practical utility in quantifying complex biomechanical phenotypes.

## 5   Limitations and Future Work

While LBM achieves efficient learning of real-world pixel dynamicity and demonstrates effectiveness in both point tracking and object tracking tasks, certain limitations persist. In point tracking applications, the collision-streaming processes remain constrained by inherent locality, leading to discontinuity issues in long-term tracking. Regarding object tracking, the current random sampling within bounding boxes exhibits vulnerability to background interference, which could potentially be mitigated by employing instance segmentation masks instead of conventional detection frameworks in future implementations. From a practical perspective, LBM shows promising potential for integration with embodied tracking tasks, where its computational efficiency and practical applicability could be further exploited through synergistic system development. Future research directions should prioritize addressing these identified constraints while exploring novel application domains.

## 6   Conclusion

This work presents the LBM, a novel framework for real-time pixel tracking. By decomposing visual objects into dynamic pixel lattices and solving motion states through collision-streaming processes, LBM achieves efficient, iteration-free tracking with the multi-layer predict-update architecture. Comprehensive evaluations on point tracking and open-world object tracking benchmarks demonstrate SOTA performance in both accuracy and efficiency. Notably, the fine-grained pixel tracking of LBM alleviates detection failure challenges inherent in object tracking applications. The lightweight design of LBM establishes new possibilities for real-world deployment in animal behavior analysis and future embodied tracking systems. LBM extends the paradigm of physics-inspired visual tracking, offering practical utility in dynamic real-world perception.

## 7   Acknowledgments

This work is supported by GRF 17201025, GRF 17200924, NSFC-RGC Joint Research Scheme N_HKU705/24, and the Natural Science Foundation of China (62461160309). This work is also supported in part by the National Natural Science Foundation of China under Grant 62173249 and the Natural Science Foundation of Shanghai under Grant 20ZR1460100.

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

# A    Detailed architecture of LBM

## A.1    LBM for Point tracking

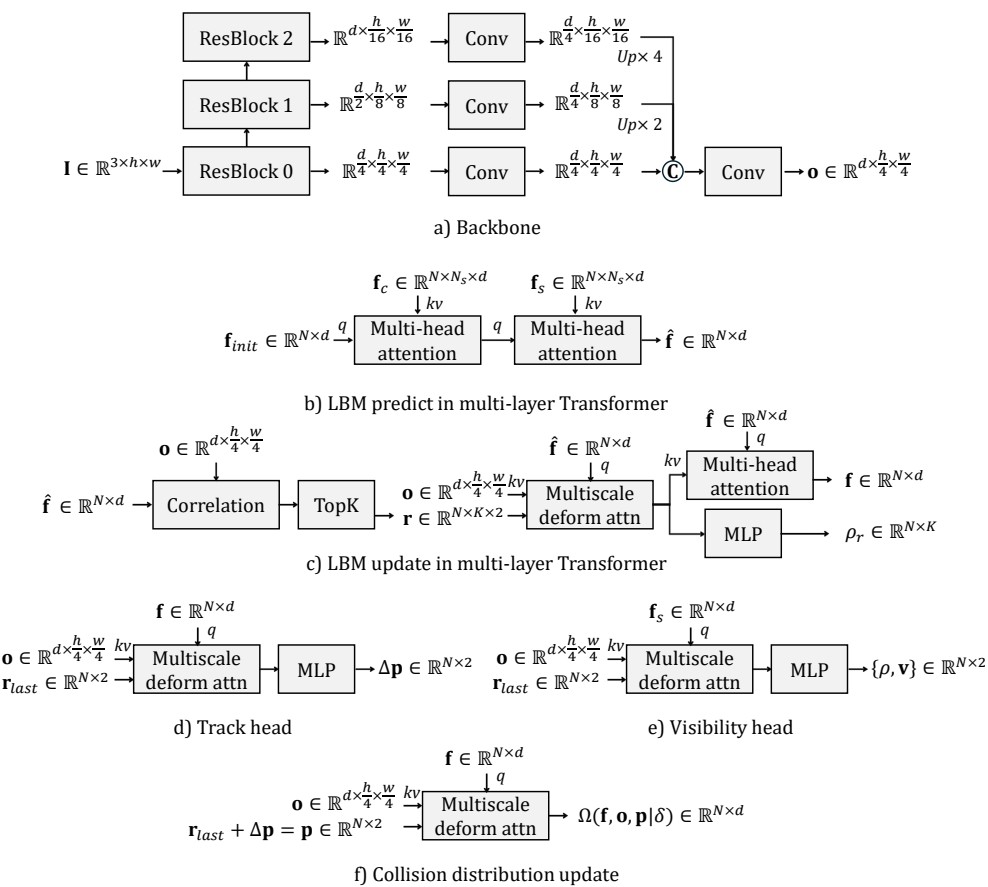

Figure 10: Detailed architecture of the proposed LBM.

**Architecture** configuration of the LBM framework is illustrated in Figure 10. Specifically, in a) *backbone* module, given image $\mathbf{I}$, the feature dimension $d$ of the output $\mathbf{o}$ is 256. In b) *LBM predict* module, $\mathbf{f}_{init}$ denotes the initial sampled distribution functions. $N$ refers to the number of points. $N_s$ denotes the number of memorized streaming distributions $\mathbf{f}_s$ and collision distributions $\mathbf{f}_c$. $\hat{\mathbf{f}}$ is the predicted query distribution. For c) *LBM Update* module, $K$ is the number of reference points for a query point, which decreases progressively as the *predict-update* layers deepen, with $K = 9$ in the first layer, $K = 1$ in the final layer, and $K = 4$ in the intermediate layers. $\rho_r$ quantifies the positional uncertainty of reference points, enforcing the spatial confinement of reference points to neighborhoods of tracked pixels. In d) *track head* and e) *visibility head* modules, the final distribution $\mathbf{f}$ and reference points $\mathbf{r}_{last}$ are derived from the last *predict-update* layer. Here, $\Delta\mathbf{p}$ represents the predicted positional offset of the pixel coordinates relative to $\mathbf{r}_{last}$, while $\mathbf{v}$ and $\rho$ are visibility and uncertainty. The streaming distributions $\mathbf{f}_s$ and collision distributions $\mathbf{f}_c$ are initialized as zero and updated as $\mathbf{f}_s = \{\mathbf{f}_i\}_{i=t-N_s}^{t-1}$ and $\mathbf{f}_c = \{\Omega(\mathbf{f}_i, \mathbf{o}_i, \mathbf{p}_i|\delta_i)\}_{i=t-N_s}^{t-1}$.

**Training loss** In Section 3.2, we discussed the fundamental loss components in LBM. Here, we provide further supplementation. The classification loss $\mathcal{L}_{cls} = \text{CE}(\mathbf{c}, \mathbf{c}_{gt}|\mathbf{v}_{gt})$ is applied at each layer of the Transformer to supervise the correlation at each level. Specifically, $\mathbf{c}_{gt}$ represents the index in the correlation map corresponding to the point's ground-truth position. The classification loss ensures that the correlation value at the ground-truth position is the highest. In regression loss $\mathcal{L}_{reg} = \text{L1}(\Delta\mathbf{p}, \Delta\mathbf{p}_{gt}|\mathbf{v}_{gt})$, $\Delta\mathbf{p}_{gt} = \mathbf{p}_{gt} - \mathbf{r}_{last}$, where $\mathbf{r}_{last}$ is the reference point from the last Transformer layer and $\mathbf{p}_{gt}$ denotes the ground-truth point location. In the confidence loss $\mathcal{L}_{conf} = \text{CE}(\sigma(\rho), \mathbb{1}[\|\mathbf{p} - \mathbf{p}_{gt}\| < 8]$, the ground-truth confidence is 1 if the mean square error

between the predicted and ground-truth points is within a threshold of 8 and the ground-truth points are not occluded; otherwise, it is 0. In addition to computing the confidence of final output points, we also supervise reference points at each layer as an auxiliary constraint to regularize their positions, *i.e.*, $\mathcal{L}_{conf,ref} = \text{CE}(\sigma(\rho_r), \mathbb{1}[\|\mathbf{r} - \mathbf{p}_{gt}\| < 8]$, where $\mathbf{r}$ is the reference point.

## A.2 Object tracking

The similarity metric for association is derived from NetTrack Zheng et al. (2024), with the cross-temporal correspondence between the $i$-th tracked instance in historical observations and the $j$-th candidate instance in current detections being formally expressed through the following formulation:

$$S_{i,j} = \min\{1, \frac{A_i}{A_j}\} \cdot \frac{N_{in}}{N_{in} + N_{out}} \quad , \tag{7}$$

where $A_i$ represents the area corresponding to the bounding box of the $i$-th instance. The scaling factor $\min\{1, \frac{A_i}{A_j}\}$ penalizes current detection instances with larger bounding areas, as expansive regions exhibit higher probabilities of containing more tracked pixels. In LBM, the similarity metric undergoes reweighting through multiplicative integration of detection scores and categorical labels. Specifically, the detection score $s_j$ and label $l_j$ of current $j$-th instances is considered:

$$S_{i,j} = s_j \cdot (0.5 + 0.5\delta_{l_i,l_j}) \cdot \min\{1, \frac{A_i}{A_j}\} \cdot \frac{N_{in}}{N_{in} + N_{out}} \quad , \tag{8}$$

where $\delta_{a,b}$ is the Kronecker delta function, defined such that $\delta_{l_i,l_j} = 1$ if $l_i$ and $l_j$ are equal, and $\delta_{l_i,l_j} = 0$ otherwise. In this context, when the label $l_i$ differs from $l_i$, a penalty weight of 0.5 is imposed. This mechanism effectively suppresses detections with ambiguous class assignments and low confidence scores by applying multiplicative attenuation to inconsistent label predictions. The final matching correspondence is determined through the normalized aggregation of the cross-frame similarity matrix $\mathbf{S} \in \mathbb{R}^{M \times N}$, computed as the arithmetic mean of bidirectional softmax-normalized distributions along both spatial dimensions (row-wise and column-wise). $M$ and $N$ respectively represent the number of tracked and newly detected instances.

## B Implementation details

**Training details** We employ the AdamW optimizer with a peak learning rate of $5 \times 10^{-4}$ and weight decay of $1 \times 10^{-5}$, implementing a cosine decay schedule with 5% linear warm-up initialization. The whole training process takes over 2 days on 4 NVIDIA H800 GPUs with 4 batches each. LBM adopts identical data augmentation strategies as CoTracker Karaev et al. (2024b), processing input images at 384×512 resolution while sampling 256 points per batch.

**Evaluation datasets** TAP-Vid DAVIS comprises 30 real-world videos sourced from the DAVIS dataset; TAP-Vid Kinetics contains 1,184 challenging real-world videos; and RoboTAP Vecerik et al. (2024) consists of 265 real-world robotic videos. Open-world object tracking datasets include: TAO Dave et al. (2020) validation set, containing 988 videos spanning 330 object categories annotated at 1 frame per second; BFT Zheng et al. (2024) test set, comprising 36 videos featuring highly dynamic avian objects; and OVT-B Liang and Han (2024), a large-scale open-world object tracking benchmark encompassing 1,973 videos with 1,048 object categories.

**Evaluation metrics** AJ serves as a comprehensive metric quantifying both position precision of predicted positions and the accuracy of visibility predictions. $\delta_{avg}^x$ evaluates the position precision of visible points by calculating the average proportion of predicted positions falling within specified thresholds (1, 2, 4, 8, 16 pixels) relative to ground-truth positions. OA specifically quantifies the accuracy of visibility predictions for occluded states. The evaluation metrics for open-world object tracking include TETA Li et al. (2022) under the open-vocabulary setting, a comprehensive metric assessing association accuracy (AssA), localization accuracy (LocA), and classification accuracy (ClsA), with rare categories in the LVIS Gupta et al. (2019) dataset designated as *novel* and the remaining categories categorized as *base*; OWTA Liu et al. (2022), a holistic evaluation metric integrating open-world object detection recall (D. Re.) and association accuracy (A. Acc.). Specifically, TETA is used for validation on TAO and OVT-B benchmarks, and OWTA is evaluated on BFT benchmark.

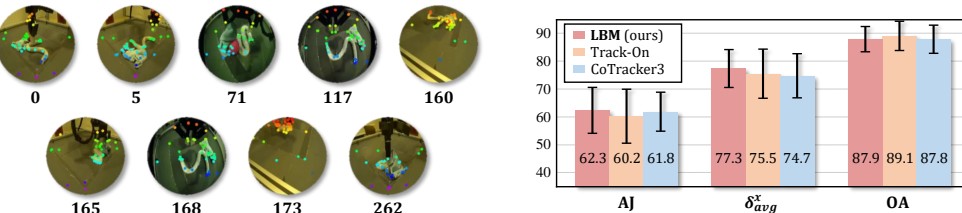

Figure 12: **Ablation on tracking manipulation of deformable objects**. A subset of RoboTAP for manipulating deformable objects was selected with video IDs. In the comparative analysis, LBM demonstrated superior tracking capabilities.

Table 8: **Ablation on the fine-grained similarity** on TAO validation benchmark. The combination of the label penalty term and detection score weight achieves the best performance.

| Label | Score | All | | | | Base | | | | Novel | | | |
|---|---|---|---|---|---|---|---|---|---|---|---|---|---|
| | | TETA | LocA | AssocA | ClsA | TETA | LocA | AssocA | ClsA | TETA | LocA | AssocA | ClsA |
| ✓ | ✓ | 45.3 | 70.0 | 32.4 | 33.4 | 46.5 | 69.9 | 33.2 | 36.4 | 36.1 | 70.8 | 26.2 | 11.4 |
| ✓ | ✗ | 45.2 | 69.8 | 32.5 | 33.3 | 46.4 | 69.7 | 33.4 | 36.3 | 35.9 | 70.8 | 25.5 | 11.4 |
| ✗ | ✓ | 44.3 | 69.6 | 29.8 | 33.4 | 45.5 | 69.5 | 30.7 | 36.3 | 34.9 | 70.1 | 23.3 | 11.3 |
| ✗ | ✗ | 44.6 | 69.4 | 31.1 | 33.4 | 45.9 | 69.2 | 32.0 | 36.4 | 35.7 | 71.0 | 24.6 | 11.4 |

## C   Detailed ablation study

**Ablation on the number of active dynamic** neighbors on TAP-Vid DAVIS is illustrated in Figure 11. Enhanced active neighbor participation improves the accuracy of collision process, thereby optimizing LBM's spatial localization precision for target pixels. This enhancement manifests through three computational phases: 1) collision distribution acquisition, 2) collision attention, and 3) inference of tracking and visibility heads. Empirical observations indicate negligible GPU memory overhead and inference speed degradation with increased neighbor counts. Given the plateau effect observed in precision gains beyond 7 neighbors, LBM establishes 9 neighbors as the optimal configuration.

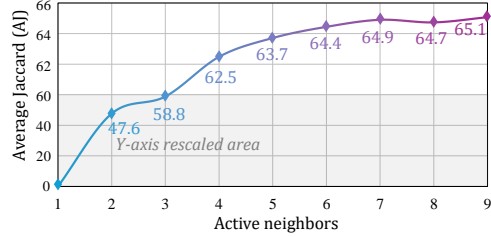

Figure 11: **Ablation on the number of active neighbors**. For enhanced visualization clarity, the segment of the Y-axis below 60 has been rescaled with a factor of 0.1. More active neighbors bring better performance.

**Ablation on tracking deformable objects in manipulation** is shown in Figure 12. A subset comprising 9 data entries is extracted from RoboTAP dataset, specifically focusing on manipulation tasks involving highly deformable objects such as ropes and fabrics. This selection explicitly excluded moderately deformable objects, including toys and shoes. A comparative analysis is conducted on this subset across three models: LBM (18 M), Track-On (49 M), and CoTracker3 (a semi-online method based on 16-frame window processing). For fairness, queries are not extended globally, but local extension is still performed in CoTracker3. Experimental results demonstrated that LBM achieved superior performance in $\delta^x_{avg}$, exhibiting enhanced localization accuracy attributed to its strengthened spatial perception capability regarding pixel-level neighborhood relationships in deformable object manipulation. This empirical investigation reveals promising potential of LBM for applications involving deformable objects.

### C.1   Ablation on object tracking

**Ablation on fine-grained similarity** on TAO validation benchmark is shown in Table 8. The label penalty term and detection score weight term in Equation 8 are comprehensively considered. As evidenced by experimental results, the label penalty term contributes a +1.4 performance gain in overall association metrics while moderately improving localization accuracy. Although isolated introduction of the detection score weight marginally enhances localization capability, it concurrently induces deterioration in association performance. Significantly, simultaneous incorporation of both

Table 9: **Ablation on AnimalTrack subset of OVT-B**. LBM shows better tracking performance against real-world object dynamicity. Best results shown in **bold**.

| Model | All | | | | Base | | | | Novel | | | |
|---|---|---|---|---|---|---|---|---|---|---|---|---|
| | TETA | LocA | AssocA | ClsA | TETA | LocA | AssocA | ClsA | TETA | LocA | AssocA | ClsA |
| SORT Bewley et al. (2016) | 54.4 | **72.2** | 19.8 | **71.3** | 54.5 | **68.3** | 18.3 | **76.9** | 54.3 | **77.9** | 22.0 | 62.9 |
| ByteTrack Zhang et al. (2022) | 61.3 | 72.0 | 40.6 | **71.3** | 60.7 | 68.1 | 37.3 | **76.9** | 62.1 | 77.8 | 45.5 | 62.9 |
| OC-SORT Cao et al. (2023) | 62.1 | 71.9 | 43.3 | 71.2 | 61.4 | 67.9 | 39.4 | **76.9** | 63.3 | 77.8 | 49.1 | 62.9 |
| **LBM** (ours) | **64.3** | 70.0 | **51.6** | 71.2 | **62.1** | 66.0 | **43.6** | 76.7 | **67.5** | 76.0 | **63.6** | **63.0** |

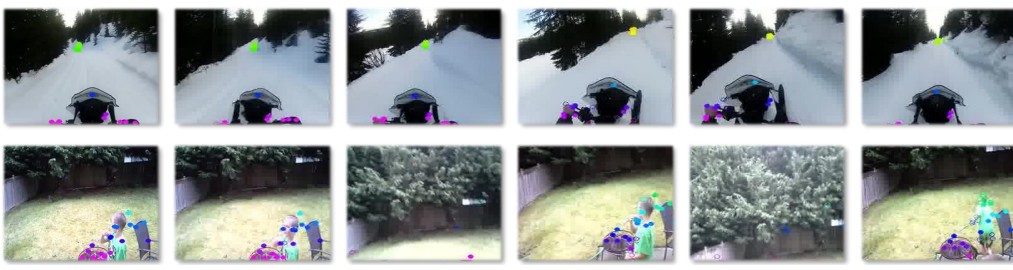

Figure 13: **Visualization of point tracking for dynamic object manipulation**. LBM shows robustness against the dynamicity of deformable objects.

label penalty and detection score weight terms synergistically elevates both LocA and AssocA. This combined approach demonstrates particularly pronounced performance improvements in novel categories, which can be attributed to the complementary mechanisms between category-aware label penalty and detection confidence weighting that effectively address both semantic alignment and spatial correspondence challenges.

**Ablation on tracking animals** is shown in Table 9. Animals typically represent highly dynamic tracking targets. In addition to demonstrating the effectiveness of LBM compared to SOTA trackers in tracking highly dynamic avian objects, as shown in Table 4, we further validated LBM's capability for animal tracking on the AnimalTrack Zhang et al. (2023) subset of OVT-B. To ensure fairness, identical detection results from the GLEE-plus detector were employed in all experiments. The results indicated that LBM achieved optimal performance in the AssocA metric compared to other trackers, with particularly notable improvements in *novel* class tracking, demonstrating a +14.5 gain over OC-SORT. The results substantiate the effectiveness and practical utility of LBM for real-world dynamic object tracking scenarios.

## D Comprehensive visualization

### D.1 Visualization of point tracking

The visualization results of the point tracking task are presented in two distinct components: the robotic manipulation scenario involving deformable objects from RoboTAP is illustrated in Figure 13, while dynamic scenes from TAP-Vid Kinetics are demonstrated in Figure 14. Benefiting from the learned collision and streaming processes, LBM maintains robust tracking performance even for

Figure 14: **Visualization of point tracking in dynamic scenes**. LBM demonstrates adaptability to dynamic environments, including scenarios involving rapid motion and viewpoint transformations.

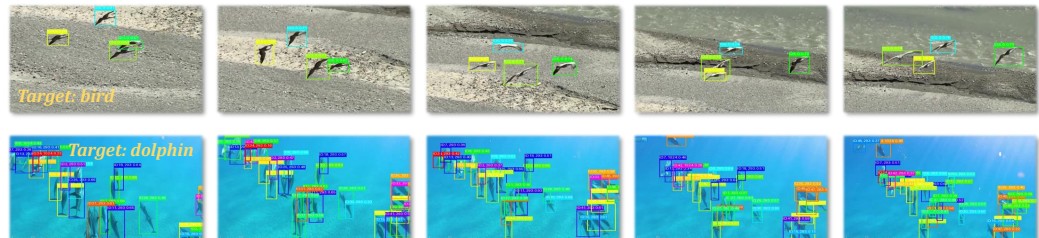

Figure 15: **Visualization of object tracking for dynamic animals**. LBM exhibits robustness against object dynamicity, such as rapid motion, deformation, similar objects, and occlusion.

highly deformable flexible objects. In dynamic environments characterized by rapid motion, such as first-person skiing scenarios with intense movement or situations involving repeated viewpoint variations, LBM exhibits remarkable adaptability. Notably, when the viewpoint temporarily loses and subsequently reacquires the target, LBM can precisely relocate tracked pixels through memorized streaming and collision distribution patterns, demonstrating exceptional robustness.

### D.2 Visualization of object tracking

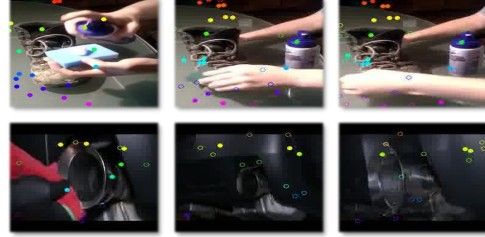

Figure 16: **Failure cases of LBM**. Tracking failures can be attributed to the uniformity in target appearance characteristics and discontinuous fragments in videos.

The visualization results of the target tracking task, as shown in Figure 15, are derived from the BFT and OVT-B datasets, respectively. Birds and dolphins, as highly dynamic targets, pose challenges including rapid motion, deformation, similar targets, and occlusion. Benefiting from the dynamic pixel management mechanism, the LBM exhibits robustness in tracking real-world dynamic objects.

## E   Failure cases and potential solutions

**Failure cases**  Figure 16 illustrates failure cases of LBM in point tracking tasks. In the first video sample, point tracking failures on the desktop surface occur due to its uniform appearance in both color and texture, revealing the localized nature of collision and streaming processes. In the second scenario, discontinuities arising from the composition of multiple spliced video segments result in observable point drift phenomena. For visual object tracking, the primary limitations persist in two aspects: 1) the random sampling within target bounding boxes exhibits inherent susceptibility to background interference contamination; 2) despite maintaining fine-grained pixel-level tracking fidelity during detection failure scenarios, the framework lacks effective mechanisms for holistic tracking state recovery at the object level, as shown in Figure 8.

**Potential solutions**  In response to the limitations inherent in LBM, several potential solutions have been enumerated as follows:

- Explicit temporal continuity mechanisms. While streaming distribution of pixels preserves temporal feature learning, the correlation for reference point acquisition in online tracking scenarios exhibits inadequate exploitation of historical positional contexts. Therefore, explicit modeling of pixel trajectory persistence emerges as a critical enhancement opportunity.

- Global semantic context augmentation. For discontinuous video sequence tracking, the semantic state coherence of pixel-associated objects inherently governs motion pattern interpretability. Developing hierarchical architectures to extract and propagate object-level semantic embeddings could substantially improve pixel motion comprehension.

- Depth-aware constraint integration. Incorporating depth-aware constraints or semantic segmentation priors could mitigate background interference during target-specific tracking, particularly through spatial attention mechanisms that discriminatively weight foreground-background sampling probabilities.

- Holistic motion decomposition modules. Implementing motion composition layers that aggregate pixel-wise displacements into interpretable object kinematics would benefit tracking state estimation and downstream applications requiring macroscopic motion understanding.

# F    Broader impacts

The proposed LBM for real-time and online pixel tracking presents both positive and negative societal implications that merit careful consideration.

## F.1    Positive societal impacts

**Enhanced efficiency in practical applications**  LBM's lightweight design and edge-device compatibility enable real-time tracking in resource-constrained scenarios. This could benefit fields like robotics and autonomous systems.

**Scientific research advancement**  As demonstrated in zebrafish behavioral analysis, LBM's ability to reconstruct 3D trajectories of dynamic objects supports quantitative studies in biomechanics and ecology. This may accelerate discoveries in genetic research or environmental monitoring.

**Robustness against system failures**  By decomposing objects into fine-grained pixels and dynamically pruning outliers, LBM reduces reliance on detection results. This improves reliability in safety-critical applications like surveillance or disaster response.

## F.2    Negative Societal Impacts

**Privacy concerns**  The technology's capacity for persistent pixel-level tracking raises risks of misuse in unauthorized surveillance. For instance, malicious actors could exploit LBM to track individuals across video feeds without consent.

**Data bias**  If LBM is finetuned on data that lacks diversity, the performance could degrade for specific demographics or scenarios, exacerbating fairness issues in deployed systems.

## F.3    Mitigation Strategies

**Strict Ethical Guidelines**  Deployment in sensitive domains, *e.g.*, public surveillance, should require transparency audits and opt-in consent mechanisms.

**Bias mitigation**  Actively curate diverse training data spanning varied motion dynamics and environmental contexts to minimize performance disparities.

