# OpenReview forum: "Lattice Boltzmann Model for Learning Real-World Pixel Dynamicity"
_NeurIPS.cc/2025/Conference — NeurIPS 2025 poster_

### Official Review · Reviewer_TFHM · 2025-06-30

**Clarity:** 3
**Significance:** 3
**Originality:** 4
**Rating:** 5
**Confidence:** 3

**Summary:**

This paper proposes a Lattice Boltzmann Model for real-time pixel tracking in videos. By modeling pixel dynamics through fluid-inspired collision-streaming processes, LBM estimates pixel positions and visibility using a lightweight predict-update network. The proposed method demonstrates computational efficiency by utilizing lightweight architectures, and experimental results on point tracking and object tracking benchmarks show its competitive accuracy and speed compared to both offline and online baselines. Its fine-grained pixel modeling also enables robust tracking under occlusion and detection failures, demonstrating strong potential for real-world applications.

**Questions:**

- Can the authors provide more empirical or theoretical evidence that the proposed collision-streaming mechanism, inspired by the lattice Boltzmann method, offers advantages in spatiotemporal modeling compared to standard attention-based architectures?
- For the object tracking benchmarks on TAO and OVT-B (Tables 2 and 3), do all compared methods, including LBM, use the same detection or segmentation backbone? If not, how do variations in detection quality influence the reported tracking performance?
- The method samples inlier pixels randomly within bounding boxes during object tracking. Have the authors considered or tested more robust strategies, such as using instance segmentation masks or uncertainty-guided sampling? This could potentially reduce the impact of background noise.

**Ethical Concerns:**

["NO or VERY MINOR ethics concerns only"]

**Final Justification:**

The authors' rebuttal effectively addresses my initial concerns, particularly regarding the computational motivation, generalization ability, and robustness in long-term tracking. They also demonstrate a clear understanding of the current limitations and outline promising directions for future improvement.

Overall, I remain positive about the paper’s contributions and believe it presents a novel and efficient approach for real-world object tracking.

**Limitations:**

yes

**Quality:**

4

**Strengths And Weaknesses:**

Strengths
- The paper addresses critical practical challenges of existing point and object tracking methods, particularly for real-time and low-latency applications on edge devices. The proposed approach introduces a novel physics-inspired framework by adapting the lattice Boltzmann method to pixel-level tracking, offering a novel perspective on spatiotemporal modeling in vision tasks.
- The architecture is thoughtfully designed for efficiency, using a multi-layer predict-update mechanism and lightweight visual encoders. The reported hardware deployment and runtime evaluations on edge platforms further validate its suitability for real-world scenarios.
- The experimental evaluation is comprehensive, covering point and object tracking benchmarks, ablation studies, and hardware tests. The comparisons against both online and offline baselines are thorough and convincingly support the claims of robustness, efficiency, and accuracy.
- The paper provides a clear discussion of its limitations, including issues related to the locality of collision-streaming processes and sensitivity to background pixels in random sampling. It also reflects on failure cases and potential solutions in the supplementary, demonstrating a comprehensive presentation.

Weakness:
- While the model is inspired by the lattice Boltzmann method, the paper provides only a loose analogy to fluid dynamics without formal theoretical justification. It lacks deeper analysis on convergence properties, stability, or potential benefits over purely attention-based models in terms of tracking accuracy or robustness.
- Although the proposed method achieves strong efficiency, its improvements in tracking accuracy over existing online methods (e.g., Track-On) are marginal (e.g., only a 0.1 increase in AJ on TAP-Vid).
- While the method is evaluated on standard datasets, its limitations under prolonged occlusions or long-horizon tracking are only briefly mentioned. Additional experiments or analysis on more challenging scenarios could strengthen the claims of robustness.

---

> ### Author Rebuttal · Authors · 2025-07-29
>
> Thank you for your positive comments regarding the computational efficiency, novelty, structural rationality of the LBM model, as well as its practical deployment and evaluation. We also appreciate your questions and suggestions to make this work better. To systematically address your concerns, we have compiled the following responses:
>
> > ### **Q1.** Can you provide more theoretical evidence that the collision-streaming mechanism offers advantages compared to standard attention-based architectures?
>
> **A1.1** Thank you for your advice on explaining the architectural advantages of the collision-streaming mechanism. The primary motivation behind LBM lies in the computational **efficiency**, especially in the collision-streaming mechanism compared to pure attention architectures.
>
> **A1.2** Taking the collision process in the lattice Boltzmann method as an instance, our approach considers modeling relationships between pixels and their dynamic local neighborhoods, rather than performing global cross-attention directly with aggregated visual embeddings.
>
> **A1.3** The collision procedure of the lattice Boltzmann method can be formally represented as:
>
> $f_{c,i} = \Omega_i(\mathbf{x} - \mathbf{c}_i\Delta{t}, t-\Delta t)\quad,$
>
> where $f_{c,i}$ means the collision term caused by the neighbor from $i$-th direction, $\Omega$ denotes the collision operator, $\mathbf{x}$ denotes the position, $\mathbf{c}$ is the direction, $t$ is the current time, and $\Delta t$ is the time interval.
> Within the LBM framework, we employ deformable attention [1] mechanisms as $\Omega$ to construct learnable dynamic neighborhoods for tracked points, with the standard visualization of this process provided in Figure 7. The collision distributions $\mathbf{f}_c$ can be mathematically formulated as:
>
> $\lbrace\Omega(\hat{\mathbf{f_i}}, \mathbf{o_i}, \mathbf{p_i} | \delta_\mathbf{i})\rbrace_{i=t-N_c}^{t-1}\quad,$
>
> where we extend the collision process from a single time step to $N_c$ steps, and multiple direction collision is merged. $\mathbf{o}$ denotes visual embeddings, $\mathbf{p}$ denotes the predicted point positions, $\delta$ is a learned dynamic neighborhood. Specifically, this procedure initially predicts $M$ candidate positional offsets to identify valuable neighboring pixels, samples corresponding neighbor embeddings from visual embeddings $\mathbf{o} \in \mathbb{R}^{h\times w \times d}$, and subsequently applies attention weighting between the pixel distribution $\hat{\mathbf{f}}$ and sampled neighbor embeddings to derive the collision distribution. The space complexity characterizing this computational procedure is formulated as $O(NMd^2)$, where $N$ denotes the number of tracked points, and $d$ corresponds to the feature dimensionality.
>
> **A1.4** If we performed global cross-attention directly between particle distributions and aggregated visual features as:
>
> $\lbrace\text{CrossAttn}(\mathbf{p_i}, \mathbf{o_i}) \rbrace_{i=t-N_c}^{t-1}\quad,$
>
> the space complexity is $O(Nhwd^2)$. Since the sampled neighbor count $M$ is significantly smaller than total $hw$ spatial positions within visual embeddings, the computational complexity of pure attention mechanisms is substantially higher than that achieved in the LBM framework.
>
> **A1.5** We sincerely appreciate your valuable suggestions and will ensure that the theoretical comparison of frameworks is presented with enhanced conceptual clarity in the revised manuscript.
>
> > ### **Q2.** How do variations in detection quality influence the reported object tracking performance?
>
> **A2.1** We appreciate your comments on the details of object tracking evaluation. Given the differing approaches among object tracking schemes, the detection results employed are typically non-uniform across methodologies. Consequently, all comparative evaluation results were derived from their respective official implementations. Considering that GLEE [2] serves as a general image and video foundation model, GLEE-generated detections were used in evaluating LBM across our benchmark assessments.
>
> **A2.2** Higher-quality detection results typically yield superior tracking performance, especially in localization accuracy (LocA). This is particularly relevant for open-world object tracking, which faces increased complexity due to challenging object categories. This phenomenon has driven tracking approaches to employ stronger detectors. For instance, MASA [3] has demonstrated the advantage of utilizing Detic [4] over alternative methods like SAM [5] and Grounding DINO [6].
>
> **A2.3** Nevertheless, the robustness of the association method in tracking remains critically important, particularly concerning the improvement of association accuracy (AssocA). Without training on domain-specific data on object tracking, the proposed LBM demonstrates the strongest overall AssocA among training-free methods on the TAO and OVT-B benchmarks, confirming its strong generalizability for real-world object tracking deployment scenarios.
>
> **A2.4** We appreciate your recommendation to clarify the impact of detection quality. We shall provide a more detailed discussion addressing this point in the revised manuscript.
>
> > ### **Q3.** Will segmentation masks or uncertainty-guided sampling benefit tracking performance?
>
> **A3.1** We appreciate your suggestion regarding the implementation of more effective point sampling schemes for object tracking. As you rightly noted, random sampling can indeed introduce background noise compared to alternative strategies like segmentation masks or uncertainty-guided sampling. To address this limitation, we have introduced a dynamic point management mechanism: LBM systematically prunes tracking points that persistently fall outside the target bounding box for a defined duration. Concurrently, it replenishes the depleted point set by sampling a corresponding number of new points within the current bounding box.
>
> **A3.2** The adoption of random sampling is motivated by our intention to employ a relaxed target prior (*e.g.*, bounding box), thereby validating the generalizability and robustness of LBM for real-world object tracking tasks.
>
> **A3.3** We acknowledge your suggestion and will incorporate more effective sampling strategies in subsequent generalization experiments to mitigate background interference during tracking, thereby further enhancing tracking performance gains.
>
> > ### **Q4.** How does LBM perform on longer-term benchmarks, which are more challenging?
>
> **A4.1** Thank you for your valuable suggestions on evaluating LBM on longer-term point tracking benchmarks. Following this suggestion, we evaluated LBM on the longer-term point tracking benchmarks, *i.e.*, ​Dynamic Replica​ and ​PointOdyssey. Dynamic Replica is a benchmark designed for 3D reconstruction and comprises 20 sequences of 300 frames each. ​PointOdyssey​ features 12 videos containing thousands of frames (up to 4,325). Consistent with the TAP-Vid benchmark, position precision $\delta_{avg}^{vis}$ of visible ​points​ is adopted as the evaluation metric. The results are shown as follows, where LBM reaches the state-of-the-art performance, especially on the long-term PointOdyssey benchmark.
>
> | Model        | Dynamic Replica | PointOdyssey |
> |--------------|:-------------------------------:|:-----------------------------:|
> | PIPs++       | 64.0                           | 32.4                          |
> | TAPIR        | 66.1                           |          -                     |
> | CoTracker    | 68.9                           | 32.7                          |
> | TAPTR        | 69.5                           |               -                |
> | LocoTrack    | 71.4                           |             -                  |
> | BootsTAPIR   | 69.0                           |             -                  |
> | CoTracker3   | 73.3                           |               -                |
> | Track-On     | **73.6**                       | 38.1                          |
> | LBM (Ours)   | 71.5                           | **41.9**                      |
>
> **A4.2** Your constructive suggestions have contributed to significant refinements in this work. In the subsequent phase, we shall conduct an in-depth investigation at both the model architecture and dataset levels to enhance robustness in long-term tracking tasks.
>
> > ### **Q5.** Why are the improvements of LBM in tracking accuracy over existing online methods marginal?
>
> **A5.1** We appreciate your insights​ regarding the performance-speed tradeoff. While LBM exhibits ​marginal improvements​ compared to recent SOTA works (*e.g.*, Track-On), it demonstrates ​superior computational efficiency​ as you noted. This aligns with our fundamental motivation: ​achieving real-time online tracking through fluid dynamics-inspired computation.
>
> **A5.2** We embrace your constructive suggestion​ and will ​diligently investigate​ strategies for ​enhancing model precision. The next effort will specifically focus on performance improvement in complex dynamic environments.
>
> > ### **Summary**
>
> We once again express our gratitude for your recognition and shall refine our work following your valuable feedback. We hope that this response adequately addresses your concerns. Should any further questions arise, we stand ready to actively engage in discussion throughout the rebuttal and discussion period.
>
> ```
> [1] Zhu, Xizhou, et al. "Deformable DETR: Deformable Transformers for End-to-End Object Detection." ICLR. 2021.
>
> [2] Wu, Junfeng, et al. "General Object Foundation Model for Images and Videos at Scale." CVPR. 2024.
>
> [3] Li, Siyuan, et al. "Matching anything by segmenting anything." CVPR. 2024.
>
> [4] Zhou, Xingyi, et al. "Detecting Twenty-Thousand Classes Using Image-Level Supervision." ECCV. 2022.
>
> [5] Kirillov, Alexander, et al. "Segment Anything." ICCV. 2023.
>
> [6] Liu, Shilong, et al. "Grounding DINO: Marrying DNO with Grounded Pre-Training for Open-Set Object Detection." ECCV. 2024.
> ```

---

> > ### Comment · Reviewer_TFHM · 2025-08-05
> >
> > I thank the authors for their detailed and thoughtful rebuttal. It effectively addresses my initial concerns, particularly regarding the computational motivation, generalization ability, and robustness in long-term tracking. The authors demonstrate a clear understanding of the current limitations and outline promising directions for future improvement.
> >
> > Overall, I remain positive about the paper’s contributions and believe it presents a novel and efficient approach for real-world object tracking.

---

> > > ### Author Response · Authors · 2025-08-05
> > > **Thank you for your positive feedback and valuable insights**
> > >
> > > We are glad to have addressed your concerns. We reiterate our appreciation for your valuable suggestions regarding the motivation, generalizability, and long-term tracking performance aspects. Your recommendations have substantially contributed to the refinement of this work. We will revise the manuscript accordingly to make this work better.

---

### Official Review · Reviewer_Ycni · 2025-07-01

**Clarity:** 3
**Significance:** 3
**Originality:** 3
**Rating:** 5
**Confidence:** 5

**Summary:**

This paper proposes using a Lattice Boltzmann Model to model the dynamics of pixels in videos, thereby enabling point tracking and object tracking tasks. Specifically, the authors use a predict-update network to estimate the positions and visibility of pixels. The predict stage simulates particle collisions, while the update stage refines the estimation using online features. Experimental results demonstrate the effectiveness of the proposed method, and the authors conduct extensive ablation studies to validate the contribution of each module.

**Questions:**

How does the proposed method perform on traditional object tracking datasets such as GOT10k, LaSOT, and TrackingNet? Is its performance consistent with that on TAO, OVT-B, and BFT?

**Ethical Concerns:**

["NO or VERY MINOR ethics concerns only"]

**Final Justification:**

The idea of using Lattice Boltzmann Model to model the dynamics of pixels in videos is interesting and novel. The experimental results support the effective of the proposed method. The authors carefully addressed my concerns on the motivation of using a shallow CNN for feature extraction and a Transformer for the Predictor and Updater modules. Therefore, I recommend accepting it.

**Limitations:**

The authors have already clearly listed the limitations in the main paper.

**Quality:**

4

**Strengths And Weaknesses:**

### Strengths
1. The use of a Lattice Boltzmann Model to model the dynamics of pixels in videos proposed by the authors is novel and effective.
2. The proposed method has significant advantages over current SOTA methods in terms of the number of parameters and running speed.
3. The proposed method can perform both point tracking and object tracking, and its generalization performance on object tracking is significantly higher than that of other methods.
### Weaknesses
1. The authors use a relatively small network (ResNet18) for feature extraction. Have they tried using deeper or larger networks? Besides the increased computational cost, would this also improve performance?
2. What is the motivation for using a transformer architecture in the Predictor and Updater modules? What would happen if CNNs were used instead?

---

> ### Author Rebuttal · Authors · 2025-07-29
>
> We sincerely appreciate your positive feedback regarding the novelty, efficiency, and generalization capabilities of the LBM. We also thank you for raising inquiries concerning the use of larger models, Transformer architectures, and the performance on traditional object tracking benchmarks. Your concerns are addressed as follows, with structured responses provided accordingly.
>
> > ### **Q1.** Will a deeper and larger feature extraction network improve performance?
>
> **A1.1** We appreciate your attention to the relationship between model parameter count and performance, along with your valuable insights.
>
> **A1.2** Theoretically, increasing parameters in the feature extraction network should enhance performance. To verify this, we replaced the ResNet18 backbone with Swin-T [1], increasing parameters from 17.8M to 27.3M. However, this modification did not yielded significant performance difference, as detailed in the table below.
>
> | Encoder | Params | Speed↑ | AJ↑ | $\delta^x_{avg}$↑ | OA↑ |
> | --- | --- | --- | --- | --- | --- |
> | Swin-T | 27.3 M | 8.9 FPS | 63.0 | 76.6 | 88.7 |
> | ResNet18 | 17.8 M | 14.3 FPS | 65.1 | 77.5 | 89.5 |
>
> **A1.3** Here is our analysis: Point tracking task relies more heavily on shallow spatial features than on deep semantic representations. This occurs because the task primarily learns spatiotemporal consistency of geometric patterns and local appearance similarity within a point's neighborhood, while remaining agnostic to object-level semantics.
>
>   We posit that such a characteristic confers strong zero-shot generalization capabilities, yet concurrently introduces a critical limitation: the lack of semantic comprehension regarding instance identity in scenes inevitably hinders long-term tracking robustness.
>
>   In future work, we will explore strategies for integrating deep semantic cues to enhance long-range point association, thereby improving the scalability of point tracking models.
>
> > ### **Q2.** What would happen if the Transformer were replaced by a CNN?
>
> **A2.1** Thank you for your inquiry regarding the architectural design of the model. Our selection of the Transformer framework was motivated principally by its two key advantages over CNNs, the capability of processing sequential information and efficiency in the proposed collision operation.
>
> **A2.2** Building upon the single-time-step streaming and collision processes of the lattice Boltzmann method, we extend the framework to multiple time steps to enhance the robustness of the particle distribution function. This extension is achieved through cross attention between the current estimated distribution $\mathbf{f}\in\mathbb{R}^{N\times{d}}$ and historical streaming distributions $\mathbf{f}_s \in \mathbb{R}^{N\times N_s \times d}$ and collision distributions $\mathbf{f}_c \in \mathbb{R}^{N\times N_c \times d}$ as:
>
> $\hat{\mathbf{f}} = \phi_{c}(\phi_{s}(\mathbf{f}, \mathbf{f}_s), \mathbf{f}_c)\quad,$
>
> where $\phi_{c}$ and $\phi_{s}$ are cross-attention modules. $\hat{\mathbf{f}}$ denotes the predicted distribution. $N$ is the number of tracked points and $d$ is the feature dimension. $N_c$ and $N_s$ respectively denote the number of time steps over which the streaming and collision distributions are recorded. Furthermore, $N_c$ and $N_s$ can be dynamically configured according to application requirements and edge device computational capabilities. However, the extended and dynamic temporal modeling remains inherently constrained by the architectural limitations of CNNs.
>
> **A2.3** Within the proposed collision operation, the interaction between the tracked points and the dynamic neighborhoods is implemented through deformable attention [2], and the collision distribution $\mathbf{f}_c$ is obtained as:
>
> $\lbrace\Omega(\hat{\mathbf{f_i}}, \mathbf{o_i}, \mathbf{p_i} | \delta_\mathbf{i})\rbrace_{\mathbf{i}=t-N_c}^{t-1}\quad,$
>
> where $\Omega$ refers to the deformable attention. $\mathbf{o}$ is the image feature, and ${\mathbf{p}}$ is the estimated positions of tracked points, which serve as reference points in the attention. $t$ is the current time step. Here are two possible solutions to use CNN to replace Transformer:
>
> - Directly using CNN to replace deformable attention, *i.e.*, the collision distribution $\mathbf{f}_c$ is formulated as:
>
>   $\lbrace\text{CNN}(\text{Sample}(\mathbf{o_i}, \mathbf{p_i})\rbrace_{\mathbf{i}=t-N_c}^{t-1},$
>
>   where sparse image features are first sampled by point positions, followed by convolution on the sampled results to generate a collision distribution. However, the size of the sampled sparse features is fixed, which hinders dynamic neighborhood perception as in Figure 7.
>
> - Alternatively, applying deformable convolution to the entire feature map before sampling to obtain the collision distribution enables dynamic neighborhoods:
>
>   $\lbrace\text{Sample}(\text{DeformCNN}(\mathbf{o_i}), \mathbf{p_i})\rbrace_{\mathbf{i}=t-N_c}^{t-1}.$
>
>   However, this approach incurs a computational complexity of $O(hwK^2d^2)$, where $h$ and $w$ are the spatial dimensions of the feature map, yielding a total of $hw$ points, $K$ is the kernel size, and $d$ is the feature dimension. This complexity is significantly higher than that of deformable attention, which operates at $O(NMd^2)$, since $N \ll hw$, where $N$ is the number of tracked points and $M$ is the number of neighbors per point.
>
>
> **A2.4** Our primary motivation for employing the Transformer lies in its capabilities for temporal modeling and enhanced computational efficiency for the collision process. For the backbone, we deliberately selected a CNN (ResNet18) over Transformer-based alternatives due to its superior efficiency in extracting shallow visual features, as discussed in Section A1.3. We appreciate your inquiry regarding the model architecture and will conduct further quantitative analysis to evaluate the feasibility of substituting CNNs for Transformers.
>
> > ### **Q3.** How does LBM perform on traditional object tracking datasets such as GOT10k, LaSOT, and TrackingNet?
>
> **A3.1** We appreciate your insights regarding the performance of LBM on traditional single-object tracking datasets. Your concerns will be addressed through two primary dimensions.
>
> **A3.2** LBM is fundamentally a pixel-tracking model devoid of object priors. Evaluations on multi-object tracking datasets (TAO, OVT-B, and BFT) follow established protocols like SORT [3] and ByteTrack [4]. which employ public detections (bounding boxes and class labels) from off-the-shelf detectors.
> However, in classical single-object tracking tasks, the bounding box of the target is directly regressed based on initial target information. Consequently, LBM requires additional object priors for assistance on datasets such as LaSOT.
>
> **A3.3** Considering that open-world multiple object tracking encompasses challenges in single object tracking across diverse scenarios and extensive categories, such as occlusions, similar objects, and deformations. Moreover, the evaluation of open-world multiple object tracking tasks includes larger-scale evaluation datasets, with totally 2,997 videos in TAO, BFT, and OVT-B. The performance of LBM in these benchmarks confirms its robustness for real-world object tracking.
>
> **A3.4** We sincerely appreciate your valuable feedback and are actively exploring the application of LBM on object tracking datasets such as LaSOT. However, due to time constraints during the rebuttal period, we will implement these tests to the fullest extent possible.
>
> > ### **Summary**
>
> We thank you again for your constructive feedback and will implement revisions accordingly. We hope these responses can address your concerns. Should further questions arise, please do not hesitate to contact us. We will be providing comprehensive responses to all inquiries during both the Rebuttal and Discussion periods.
>
> ```
> [1] Liu, Ze, et al. "Swin Transformer: Hierarchical Vision Transformer Using Shifted Windows." Proceedings of the IEEE/CVF international conference on computer vision. 2021.
>
> [2] Zhu, Xizhou, et al. "Deformable DETR: Deformable Transformers for End-to-End Object Detection." International Conference on Learning Representations. 2021.
>
> [3] Bewley, Alex, et al. "Simple Online and Realtime Tracking." IEEE International Conference on Image Processing. 2016.
>
> [4] Zhang, Yifu, et al. "ByteTrack: Multi-Object Tracking by Associating Every Detection Box." European conference on computer vision. 2022.
> ```

---

> > ### Comment · Reviewer_Ycni · 2025-08-05
> >
> > Thank you for the authors' response. They addressed all my concerns and I will keep my inital rating.

---

> > > ### Author Response · Authors · 2025-08-05
> > > **Thanks for your time, efforts, and recognition**
> > >
> > > We are pleased to have addressed your concerns. We extend our sincere appreciation for your recognition of LBM and your valuable feedback. We will undertake revisions to the manuscript in accordance with your suggestions to enhance the presentation of this work.

---

### Official Review · Reviewer_pfjb · 2025-07-02

**Clarity:** 3
**Significance:** 2
**Originality:** 3
**Rating:** 5
**Confidence:** 4

**Summary:**

This work introduces a novel Lattice Boltzmann Model (LBM) designed for real-time visual tracking of dynamic objects. The model decomposes visual information into dynamic pixel lattices and utilizes collision-streaming processes to analyze pixel motion effectively. It employs a multilayer predict-update network to estimate distributions of target pixels, refining estimations by managing lattice interactions during prediction and leveraging online visual features in the updating phase. The model’s performance is evaluated using benchmarks like TAP-Vid and RoboTAP, showing superior efficiency and practicality compared to traditional offline methods. LBM addresses significant challenges such as high resource consumption and latency, enhancing robustness in tracking scenarios involving occlusion and deformation.

**Questions:**

SEE WEAKNESS

**Ethical Concerns:**

["NO or VERY MINOR ethics concerns only"]

**Final Justification:**

ACCEPTED

**Limitations:**

SEE WEAKNESS

**Quality:**

3

**Strengths And Weaknesses:**

Strengthens：
1. The use of the Lattice Boltzmann Model to decompose visual information into dynamic pixel lattices is a novel approach that offers a new perspective on pixel tracking.
2.The paper effectively demonstrates LBM’s ability to perform real-time tracking tasks, which is crucial for applications in resource-constrained environments.
3. The work outlines a theoretical framework rooted in the lattice Boltzmann method for fluid simulations, bridging established physics principles with modern computational techniques, thereby enriching the understanding of pixel tracking dynamics
4.The paper presents extensive evaluations on widely recognized benchmarks like TAP-Vid and RoboTAP, demonstrating LBM’s efficiency and practicality in both point tracking and large-scale open-world object tracking tasks

Weakness
1.The performance improvements could raise concerns regarding overfitting, especially if the model is highly tailored to the benchmarks used for evaluation. More diverse datasets would provide a clearer indication of generalizability
2.The proposed model’s complexity may pose challenges in practical deployment, particularly in less controlled environments where computational resources are limited. Simplifying the architecture or providing detailed guidelines for implementation could enhance its adoption.
3.Although the model is designed for resource-constrained environments, its computational requirements could still be a barrier in extremely low-resource scenarios.

---

> ### Author Rebuttal · Authors · 2025-07-29
>
> We sincerely appreciate your recognition of the novelty, computational efficiency, and physics-inspired principles of the LBM methodology, as well as the comprehensiveness of evaluation. We are also grateful for your insightful comments regarding generalizability, deployment complexity, and performance in low-resource scenarios. We address your questions as follows.
>
> > ### **Q1**. Does the model exhibit potential overfitting risks, and what is its generalization capability on broader datasets?
>
> **A1.1** We appreciate your concerns regarding the model's generalization capability. We will address these issues from the perspective of the heterogeneity of training and evaluation data.
>
> **A1.2** The training dataset for the LBM comprises the Kubric Movi-F benchmark generated by the Kubric [1] simulator, following CoTracker [2] and TAPTR [3]. Specifically, the dataset demonstrates a 3D rigid body falling, colliding with the ground, and rebounding. The diversity of scenarios and objects in the scalable Kubric simulator empowers LBM to achieve robust generalization across extensive real-world applications.
>
> **A1.3** The evaluation for LBM encompasses benchmarks for point tracking tasks like TAPVid-DAVIS, TAPVid-Kinetics, and RoboTAP, comprising 1,479 real-world scenarios, alongside the data for object tracking tasks, including 2,997 real-world scenarios in TAO, BFT, and OVT-B. These datasets cover diverse indoor/outdoor environments and targets, including animal activities, autonomous driving, embodied manipulation, and crowd behaviors. More importantly, as the evaluation data originates from a different domain than the simulated training data, LBM's performance on the evaluation sets ​demonstrates its strong zero-shot generalization capability.
>
> **A1.4** Following your suggestion, LBM was further evaluated on the longer-term point tracking benchmarks, *i.e.*, ​Dynamic Replica​ and ​PointOdyssey. Dynamic Replica is a benchmark designed for 3D reconstruction and comprises 20 sequences of 300 frames each. ​PointOdyssey​ features 12 videos containing thousands of frames (up to 4,325). Consistent with the TAP-Vid benchmark, position precision $\delta_{avg}^{vis}$ of visible ​points​ is adopted as the evaluation metric. The results are shown as follows, where LBM reaches the state-of-the-art performance, especially on the long-term PointOdyssey benchmark.
>
> | Model        | Dynamic Replica | PointOdyssey |
> |--------------|:-------------------------------:|:-----------------------------:|
> | PIPs++       | 64.0                           | 32.4                          |
> | TAPIR        | 66.1                           |          -                     |
> | CoTracker    | 68.9                           | 32.7                          |
> | TAPTR        | 69.5                           |               -                |
> | LocoTrack    | 71.4                           |             -                  |
> | BootsTAPIR   | 69.0                           |             -                  |
> | CoTracker3   | 73.3                           |               -                |
> | Track-On     | **73.6**                       | 38.1                          |
> | LBM (Ours)   | 71.5                           | **41.9**                      |
>
> **A1.5** We are actively conducting experiments to evaluate the generalization capabilities of LBM across more benchmarks and tracking-related tasks. We shall further mitigates overfitting risks through training on more extensive and diversified datasets.
>
> > ### **Q2.** How to mitigate model complexity and deployment challenges in practical scenarios?
>
> **A2.1** We appreciate your insightful comments on model deployment challenges and valuable suggestions. To address your concerns, we will implement solutions from two aspects, focusing on edge device adaptation and comprehensive documentation.
>
> **A2.2** To further mitigate deployment complexity, we eliminated the dependency on CUDA-compiled multi-scale deformable attention from the MMCV library, substituting it with native PyTorch source code. Subsequently, the model was compiled into an ONNX format to enhance cross-platform deployment compatibility. Furthermore, FP16 quantization via TensorRT was implemented to accelerate inference on widely adopted embedded systems such as the NVIDIA Jetson series, as shown in the table below.
>
> | Platform | Model | FPS |
> | --- | --- | --- |
> | Orin NX | MMCV | 14  |
> | Orin NX | PyTorch | 7   |
> | Orin NX | TensorRT | 49  |
>
> Compared with the reported inference speed, the quantized model realizes a $\times3.5$ acceleration with 49 frames per second (FPS).
>
> **A2.3** Complementing our quantization models, the open-source repository implements comprehensive documentation for LBM inference and edge deployment. This encompasses:
>
> - Dependency installation across diverse operating systems
>
> - Conversion of open-source weights to ONNX models
>
> - Transformation into TensorRT engines
>
> - Inference demos for both ONNX and TensorRT formats
>
> - Benchmarking velocity tests across heterogeneous models and devices
>
> - Troubleshooting guides for common issues.
>
>
> All associated code, documentation artifacts, and deployment methodologies will be fully open-sourced.
>
> **A2.4** We thank your advice and will further streamline model inference and deployment pipelines to improve implementation in real-world application scenarios.
>
> > ### **Q3.** How can LBM be effectively deployed on edge devices with more constrained computational and memory resources?
>
> **A3.1** We appreciate your insightful comment on the more resource-constrained deployment. LBM benefits from its lightweight model architecture and online inference manner, exhibiting low GPU memory consumption that scales approximately linearly with increasing number of points, as shown in the table below. $N$ is the number of tracked points, the maximum consumed GPU memory is monitored, and the average memory consumption of a single point compared with $N=1$ is reported. The test platform is an NVIDIA Jetson Orin NX.
>
> | $N$ | 1   | 2   | 5   | 10  | 20  | 50  | 100 |
> | --- | --- | --- | --- | --- | --- | --- | --- |
> | Max memory (MB) | 159 | 160 | 164 | 169 | 181 | 214 | 271 |
> | Average point memory (MB) | -   | 1.00 | 1.25 | 1.11 | 1.16 | 1.12 | 1.13 |
>
> **A3.2** We further implemented LBM with FP16 quantization via TensorRT for higher inference efficiency, as discussed in A2.2.
>
> **A3.3** We will test the efficiency of LBM on edge devices with more limited resources, *e.g*., NVIDIA Jetson Nano, to further realize broader practicality.
>
> > ### **Summary**
>
> We appreciate your valuable comments regarding the model's generalizability and deployment considerations. We hope our responses have addressed your concerns. During the Rebuttal and Discussion period, we will promptly address any further questions you may have. Should your concerns be resolved, please consider reassessing the review of our work. We respectfully look forward to your reply!
>
>
> ```
> [1] Greff, Klaus, et al. "Kubric: A Scalable Dataset Generator." Proceedings of the IEEE/CVF conference on computer vision and pattern recognition. 2022.
>
> [2] Karaev, Nikita, et al. "Cotracker: It is Better to Track Together." European conference on computer vision. 2024.
>
> [3] Li, Hongyang, et al. "TAPTR: Tracking Any Point with Transformers as Detection." European Conference on Computer Vision. 2024.
> ```

---

> > ### Comment · Area_Chair_dW2K · 2025-08-05
> >
> > Dear Reviewer #pfjb, thank you for contributing your time to review this manuscript. Please note that the authors have provided a detailed rebuttal. As part of the review process, we kindly request that you review this response and confirm whether your concerns have been addressed. We appreciate your prompt attention to this mandatory step in the review process.

---

> > > ### Comment · Area_Chair_dW2K · 2025-08-07
> > >
> > > Hi Reviewer #pfjb ,
> > >
> > > Please check the author's feedback, evaluate how it addresses the concerns you raised, and discuss the rebuttal with the authors. Notice the acknowledgement is mandatory. Please do this ASAP. Thanks.
> > >
> > > Your AC.

---

> ### Comment · Reviewer_pfjb · 2025-08-07
>
> The response has clarified all my concerns， I will raise my score rating

---

> > ### Author Response · Authors · 2025-08-07
> > **We sincerely appreciate your constructive suggestions and positive feedback**
> >
> > We are pleased to have resolved your concerns. Following your suggestions regarding model generalizability and deployment, explanations addressing these aspects will be incorporated into the revised manuscript. Once again, your valuable contributions to improving this work are gratefully recognized.

---

### Official Review · Reviewer_6HeP · 2025-07-02

**Clarity:** 1
**Significance:** 3
**Originality:** 3
**Rating:** 3
**Confidence:** 3

**Summary:**

This paper proposes a new method for point tracking. The proposed method, LBM, utilizes ideas from deformable attention to construct a more efficient neural network compared to prior works. Extensive experiments are provided to highlight the SOTA performance of the proposed LBM compared to online tracking methods. LBM seems particularly effective in resource constrained settings.

**Questions:**

Please see the weaknesses section for questions.

**Ethical Concerns:**

["NO or VERY MINOR ethics concerns only"]

**Final Justification:**

I appreciate the clarifications provided by the authors, and I do think the paper has novelty and technical merit. However, I choose to keep my score because I think the paper needs to undergo a major revision to improve its clarity and mathematical rigor.

**Limitations:**

Yes.

**Paper Formatting Concerns:**

None.

**Quality:**

2

**Strengths And Weaknesses:**

**Strengths:**

The proposed method provides promising efficiency gains over prior works, and therefore can be a valuable practical system for tracking. The paper also provides several interesting ablation studies that reflect the importance of some of its design choices.

**Weaknesses:**

I have the following concerns and questions about the paper:

1- The paper is not easy to understand, and lacks in mathematical rigor when presenting key components such as the collision and streaming distributions (lines 144-147). It also moves the loss computation to the appendix, which limits the clarity of the main paper. Even the explanation in the appendix lacks exact mathematical definitions for various operations.

2- The connection to the lattice Boltzmann method is not clear to me. In particular, I don’t understand what is the value (and relevance) of Equations 1-4 to the actual proposed method. Are these equations providing some performance guarantees, if so, how and what?

3- The description of the method remains on a very high level in my opinion, not discussing some important design choices such as the selection of reference points, and collision and streaming distributions.

---

> ### Author Rebuttal · Authors · 2025-07-29
>
> We appreciate your recognition of the efficiency and ablation studies, as well as your detailed comments that will help improve this work. To answer your questions more systematically, we have rearranged the questions as follows.
>
> > ### **Q1** What’s the connection between the lattice Boltzmann method and the proposed LBM? In particular, what’s the contribution of Equations 1\~4 to LBM?
>
> **A1.1** Thank you for pointing out that the clarity in the theoretical support for LBM needs improvement. Equations 3\~5 are the core pipeline of LBM, which are derived from Equations 1\~2, the discrete distribution function of the lattice Boltzmann method.
>
> **A1.2** The core inspiration from the lattice Boltzmann method lies in its high computational efficiency. During streaming and collision processes, it circumvents the inherent complexity of modeling long-range particle interactions by computing interactions only between a point and its neighbors within a defined local vicinity. The proposed LBM adopts this core computational logic, but further merges computations in the neighborhood and makes temporal extension.
>
> **A1.3** Specifically, as shown in Equation 1, to update the distribution function over time, the Lattice Boltzmann Method computes the directional distribution functions of fluid particles within a single time step through streaming and collision processes. For example, in a two-dimensional scenario, an individual lattice interacts with 9 fixed neighbors, with its state represented by the distribution functions denoted as $\mathbf{f}=\lbrace{f_i}\rbrace_{i=0}^{8} \in \mathbb{R}^9$. In the $i$-th direction, the streaming term is the distribution function of the previous time step, and the collision term describes the collision between the particle and its neighbors.
>
> **A1.4** In this work, we first represent the multi-direction distribution functions as a $d$-dimensional visual embedding $\mathbf{f} \in \mathbb{R}^d$, as formulated in Equation 3. The collision process in multiple directions is combined into a **dynamic neighborhood** $\delta$ for efficiency. Furthermore, LBM extends the **temporal context** of the lattice Boltzmann method, extending the evolution of a single time step to learning the streaming and collision processes of the past $N_s$ and $N_c$ time steps. This extension integrates the temporal evolution of distribution functions over multiple time steps, ​thereby enabling the robust modeling of long-range temporal dependencies inherent in complex dynamic scenes. In this process, the distribution functions of the previous $N_s$ time steps constitute the streaming distribution $\mathbf{f}_s \in \mathbb{R}^{N_s \times d}$, and the collision results of the previous $N_c$ time steps establish the collision distribution, denoted as $\mathbf{f}_c \in \mathbb{R}^{N_c \times d}$. We implement two distinct cross-attention modules to interact between the current distribution function and both the streaming distribution $\mathbf{f}_s$ and collision distribution $\mathbf{f}_c$.
>
> **A1.5** In the revised manuscript, based on the single-time-step form in Equation 3, we will further present the LBM formulation with extended temporal context, including learning from streaming distributions $\mathbf{f}_s$ and collision distributions $\mathbf{f}_c$ as:
>
> $\hat{\mathbf{f}} = \phi_{c} (\phi_{s} (\mathbf{f}, \mathbf{f}_s), \mathbf{f}_c) \quad ,$
>
> where $\hat{\mathbf{f}}$ is the predicted distribution function, $\phi_{c}$ and $\phi_{s}$ are cross-attention modules. The acquisition of $\mathbf{f}_s$ and $\mathbf{f}_c$ will be discussed in Q2\&A2.
>
> > ### **Q2** How to understand the key components, such as the collision and streaming distributions?
>
> **A2.1** We acknowledge the need for enhanced mathematical rigor in our derivations, as mentioned in your feedback. The streaming and collision distribution is obtained as follows: we compose the final distribution function output by the Transformer in the previous $N_s$ time steps into the streaming distribution $\mathbf{f}_s$:
>
> $\lbrace\hat{\mathbf{f_i}}\rbrace_{\mathbf{i}=t-N_s}^{t-1}\quad,$
>
> where $\hat{\mathbf{f}}$ denotes the final distribution function acquired from the Transformer, and $t$ refers to the current time step. We compose the interaction result in the dynamic neighbors into the collision distribution $\mathbf{f}_c$:
>
> $\lbrace\Omega(\hat{\mathbf{f_i}}, \mathbf{o_i}, \mathbf{p_i} | \delta_\mathbf{i})\rbrace_{\mathbf{i}=t-N_c}^{t-1}\quad,$
>
> where $\Omega$ is the collision operator in Equation 3, implemented as multi-scale deformable attention [1]. $\mathbf{o}$ denotes the image feature and $\mathbf{p}$ is the estimated position of tracked points, which serves as reference points in deformable attention. $\delta$ is the learned dynamic neighborhood.
>
> **A2.2** The distinction from the lattice Boltzmann method in Equation 1 lies in our employment of multi-time-step streaming terms and collision terms. This temporal extension facilitates the establishment of longer-term temporal dependencies, thereby enabling the robust handling of challenging dynamic scene variations​, such as deformations and illumination changes, ultimately enhancing temporal robustness.​​
>
> > ### **Q3** How to explain the mathematical definitions of operations in LBM?
>
> **A3.1**  We are grateful for your recommendations on refining the mathematical definitions of the LBM operations. The detailed explanation of LBM operations is outlined as follows:
> - Visual encoding: $\mathbf{o} = \mathcal{E}(\mathbf{I})$, where $\mathbf{o}\in\mathbb{R}^{h\times{w}\times{d}}$ is the visual features with $d$ dimensions. The detailed architecture of visual coder $\mathcal{E}$ is shown in Figure 10-a, which includes 3 ResBlocks from ResNet18 and 3 projection layers. $\mathbf{I} \in \mathbb{R}^{3\times{H}\times{W}}$ is the input image. The stride of the output feature $\mathbf{o}$ is 4, *i.e.*, $h=\frac{H}{4}$ and $w=\frac{W}{4}$.
> - Initial sampling of pixel distribution: given $N$ point queries $\mathbf{q}\in\mathbb{R}^{N\times2}$, the initial pixel distributions of the queries is acquired by bilinear sampling from the visual features $\mathbf{o}$, which is formulated as: $\mathbf{f}_{init}=\text{BilinearSample}(\mathbf{o}, \mathbf{q}) \in \mathbb{R}^{N\times d}$.
> - Predict current distribution functions: the pixel distribution function evolves temporally, with the current time step being predicted from temporal context with streaming distributions $\mathbf{f}_s \in \mathbb{R}^{N \times N_s\times d}$ and collision distributions $\mathbf{f}_c \in \mathbb{R}^{N \times N_c\times d}$, which are discussed in A1.3 and A2.1 in detail.
>
>   This process can be mathematically formalized as follows: $\hat{\mathbf{f}} = \phi_{c}(\phi_{s}(\mathbf{f}, \mathbf{f}_s), \mathbf{f}_c)$, where $\mathbf{f}$ is the input distribution functions.
> - Update distribution functions with image: as illustrated in Figure 3, the predicted distributions undergo correlation operations with current image features $\mathbf{o}_t$ at time step $t$, subsequently triggering the selection of top-$K$ response points as reference points $\mathbf{r}\in\mathbb{R}^{N\times K \times 2}$. These points then guide the updating of distribution functions through deformable attention modules utilizing image features. This process can be formulated as: ${\mathbf{r}}= \text{TopK}(\text{SoftArgmax}(\langle {\mathbf{f}}, \mathbf{o} \rangle))$, where $\langle,\rangle$ denotes correlation. And the updated distribution functions are: $\mathbf{f} = \text{MSDeformAttn}(\hat{\mathbf{f}}, \mathbf{o}, {\mathbf{r}})$.
> - Multi-layer predict-update Transformer: we employ a multi-layer Transformer architecture, where each layer implements sequential prediction and update processes. As discussed in Section 3.3, the parameter $K$ in the top-$K$ operations diminishes as the layers get deeper, and $K=1$ in the last layer denotes the final reference points $\mathbf{r}_{last}\in\mathbb{R}^{N\times2}$. The final output distribution functions $\mathbf{f}$ are fed into the track head and visibility head to predict the point position and visibility.
>
>   Both the track head and visibility head consist of a deformable attention module and an MLP layer. The track head predicts the offset of tracked points from the final reference points $\mathbf{r}_{last}$ as:
>
>    $\Delta \mathbf{p}=\mathcal{H_{track}}(\mathbf{f}, \mathbf{o}, \mathbf{r}_{last}) \in \mathbb{R}^{N\times2}$.
>
>   And the visibility head estimates the visibility and confidence of tracked points as:
>
>   ${\rho}=\mathcal{H_{vis}}(\mathbf{f}, \mathbf{o}, \mathbf{r}_{last}) \in \mathbb{R}^{N\times2}$.
>
>   The current estimated position $\mathbf{p}=\Delta {\mathbf{p}}+\mathbf{r}_{last}$.
>
> - Acquisition of streaming distributions $\mathbf{f}_s$ and collision distributions $\mathbf{f}_c$ are discussed in A2.1 in detail.
>
> **A3.2** We thank your advice and will systematically formalize the LBM workflow with mathematical rigor in the main text to enhance readability.
>
> > ### **Q4**  Why is loss computation moved to the Appendix?
>
> **A4.1**  We sincerely appreciate your feedback regarding the explanation of the loss function. We discuss the loss function in the Appendix primarily due to space constraints in the original manuscript. We will incorporate essential technical elaborations on the loss functions within the revised body text for better clarity.
>
> > ### **Summary**
>
> Thank you again for helping us improve the paper. We hope our response can resolve your concerns. Please let us know if you have any further questions. We will be actively available during the rebuttal and discussion period. If you feel your concerns are addressed, please consider reevaluating our work. Looking forward to hearing from you!
>
> ```
> [1] Zhu, Xizhou, et al. "Deformable DETR: Deformable Transformers for End-to-End Object Detection." International Conference on Learning Representations. 2021.
> ```

---

### Note · Authors · 2025-08-12

We sincerely thank every reviewer and the chair for their time and effort. We greatly appreciate the reviewers' recognition of our **novelty**, **motivation**, **efficiency**, **practicality**, and **comprehensive evaluation**. We would also like to express our sincere appreciation to the reviewers for their valuable suggestions to further improve the paper, including:
- Reviewer 6HeP‘s suggestions on clarity and theory.
- Reviewer pfjb‘s suggestions on practical deployment.
- Reviewer Ycni‘s suggestions on model design.
- Reviewer TFHM‘s suggestions on theoretical aspects and long-term tracking.

During the rebuttal phase, we did out best to addresse all reviewers' concerns and rigorously improved our work, as detailed below:

1. **Theoretical clarification (Reviewers 6HeP & TFHM):**

    The migration process from Lattice Boltzmann Method theory to the proposed LBM has been further clarified, specifically in the areas of streaming and collision processes. We mathematically elaborated on LBM's implementation, while explicitly comparing its computational efficiency against standard attention mechanisms.

2. **Generalization on long-term tracking (Reviewers TFHM & pfjb):**

    Additional evaluations were conducted on the long-term point tracking datasets Dynamic Replica and Point Odyssey. Results confirm that LBM achieves SOTA performance in long-term tracking scenarios, demonstrating robust generalization capability.

3. **Edge deployment optimization (Reviewer pfjb):**

    To further address practical deployment constraints, we implemented FP16 quantization via TensorRT on the NVIDIA Jetson Orin NX edge device. This yielded a 3.5× speedup (from 14 FPS to 49 FPS) compared to the originally reported results. Detailed open-source documentation has been organized to facilitate community adoption.

4. **Model design explanation (Reviewer Ycni):**

    Experimental studies with a deeper Swin-T backbone are presented, and we theoretically explain the reasons for its limited performance. We further justified the superiority of the proposed Transformer in LBM over CNN in temporal modeling and the collision process through complexity analysis.

We will systematically document all improvements arising from the rebuttal and discussion in the revised manuscript, accompanied by open-sourcing of the code with comprehensive documentation for community reference. We thank reviewers and the chair again for their support in this work.

---

### Decision · Program_Chairs · 2025-09-17

**Decision:**

Accept (poster)

**Comment:**

The paper was generally well-received, obtaining three accepts and one borderline reject, with reviewers highlighting its novelty, efficiency, and practical relevance. Nonetheless, Reviewer #6HeP raised significant concerns regarding clarity and mathematical rigor, citing the lack of precise definitions for the collision and streaming processes, the unclear connection to lattice Boltzmann theory, and omission of important design choices. Reviewer #pfjb acknowledged the contributions but noted risks of overfitting, deployment complexity, and limitations in very low-resource environments. Reviewer #TFHM emphasized weak theoretical grounding, marginal accuracy improvements over baselines, and insufficient evaluation of long-term tracking robustness.

In the rebuttal, the authors provided clarifications on the theoretical grounding of LBM (addressing concerns from #6HeP and #TFHM), added results on Dynamic Replica and PointOdyssey to evaluate long-term tracking (#TFHM, #pfjb), demonstrated optimized deployment with FP16 quantization on edge devices (#pfjb), and included analyses of deeper backbones and Transformer justification (#Ycni). Reviewers #pfjb, #Ycni, and #TFHM confirmed that their concerns were resolved. Reviewer #6HeP acknowledged the technical merits but maintained that the paper requires substantial rewriting to improve clarity and rigor.

After considering the paper, rebuttal, and reviewer feedback, the AC recognizes the novelty and practical value of the work and supports the majority opinion, recommending acceptance. The AC strongly encourages the authors to revise the manuscript thoroughly for the camera-ready version, particularly addressing the clarity and rigor issues emphasized by Reviewer #6HeP.